# 4DPC²hat: Towards Dynamic Point Cloud Understanding with Failure-Aware Bootstrapping

Xindan Zhang [1 2]   Weilong Yan [3]   Yufei Shi [4]   Xuerui Qiu [5 6]
Tao He [7]   Ying Li [1 2]   Ming Li [8]   Hehe Fan [9]

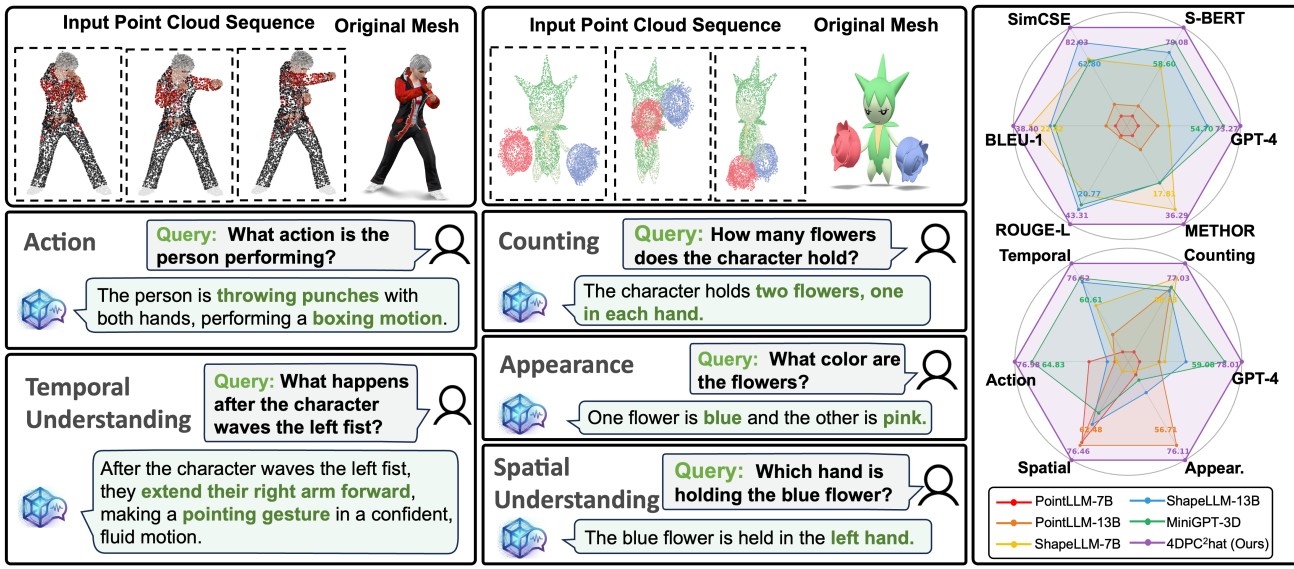

Figure 1. (a) We present 4DPC²hat, the *first* multimodal large language model specially tailored for 4D point cloud sequence understanding, which supports the answering of diverse challenging question types, from action recognition to temporal relations, etc. (b) Compared with existing state-of-the-art models, 4DPC²hat demonstrates significantly superior performance on the comprehensive metric set.

## Abstract

Point clouds provide a compact and expressive representation of 3D objects, and have recently been integrated into multimodal large language models (MLLMs). However, existing methods primarily focus on static objects, while understanding dynamic point cloud sequences remains largely unexplored. This limitation is mainly caused by the lack of large-scale cross-modal datasets and the difficulty of modeling motions in spatio-temporal contexts. To bridge this gap, we present 4DPC²hat, the *first* MLLM tailored for dynamic point cloud understanding. To this end, we construct a large-scale cross-modal dataset 4DPC²hat-200K via a meticulous two-stage pipeline consisting of topology-consistent 4D point construction and two-level captioning. The dataset contains over 44K dynamic object sequences, 700K point cloud frames, and 200K curated question–answer (QA) pairs, supporting inquiries about counting, temporal relationship, action, spatial relationship, and appearance. At the core of the framework, we introduce a Mamba-enhanced temporal reasoning MLLM to capture long-range dependencies and dynamic patterns among a point cloud sequence. Furthermore, we propose a *failure-aware* bootstrapping learning strategy that iteratively identifies model deficien-

[1]College of Computer Science and Technology, Jilin University [2]Key Laboratory of Symbolic Computation and Knowledge Engineering of Ministry of Education, Jilin University [3]National University of Singapore [4]Nanyang Technological University [5]Institute of automation, Chinese Academy of Sciences [6]Zhongguancun Academy [7]University of Electronic Science and Technology of China [8]School of Artificial Intelligence, The Chinese University of Hong Kong, Shenzhen [9]Zhejiang University. Correspondence to: Ming Li <ming.li@u.nus.edu>.

*Proceedings of the 43rd International Conference on Machine Learning*, Seoul, South Korea. PMLR 306, 2026. Copyright 2026 by the author(s).

cies and generates targeted QA supervision to continuously strengthen corresponding reasoning capabilities. Extensive experiments demonstrate that our 4DPC$^2$hat significantly improves action understanding and temporal reasoning compared with existing models, establishing a strong foundation for 4D dynamic point cloud understanding.

## 1. Introduction

Point clouds have been widely adopted in various real-world applications such as embodied AI (Wang et al., 2024b), robotics (Chen et al., 2024; Yan et al., 2026), and autonomous driving (Li et al., 2021; Arnold et al., 2019; Yan et al., 2025; Chen et al., 2026) since they offer a native, sensor-aligned representation of 3D geometry, preserving fine-grained spatial structures while remaining sparse and computationally efficient. With the rapid progress of multimodal large language models (MLLMs), recent studies have begun extending LLMs to 3D point cloud domains (Hong et al., 2023a; Guo et al., 2023; Xu et al., 2024). These advances pioneer the unification of point clouds and natural language within a shared reasoning framework, leading to substantial improvements in 3D recognition, cross-modal alignment, and interactive understanding.

Despite these advances, existing methods remain largely restricted to static point clouds (Jia et al., 2025; Azuma et al., 2022; Xu et al., 2024; Guo et al., 2023; Qi et al., 2024a; Tang et al., 2024). Their training data and model architectures are primarily designed for single-frame point clouds. In contrast, real-world perception and reasoning often require understanding dynamic point clouds, *i.e.*, sequences of point sets evolving over time, which is crucial for capturing object actions, state transitions, and complex spatio-temporal interactions (Wei et al., 2022; Aygun et al., 2021; Min et al., 2020). Without explicitly modeling temporal dynamics, current 3D MLLMs lack the temporal modeling and reasoning capabilities necessary to address these challenges.

As a result, progress in 4D point cloud understanding still faces several fundamental challenges. First, large-scale and well-aligned cross-modal datasets containing text-4D object pairs are extremely scarce. Unlike static point clouds, 4D point clouds require temporally coherent acquisition across frames, which demands accurate temporal alignment, stable object tracking, and reliable frame-to-frame correspondence, making data collection substantially more complex. Moreover, existing datasets are primarily designed for single-modal tasks such as pose estimation (Wang et al., 2020) or action classification (Deng et al., 2024), and thus lack language-centric supervision and cross-modal alignment.

Second, spatio-temporal modeling of 4D point clouds is inherently challenging. Beyond processing irregular and complex 3D structures at each time step, models must reason over temporally evolving geometry (Liu et al., 2025; Wang et al., 2022), where point distributions, object topology, and local spatial relationships change continuously. Correctly interpreting object actions or interactions, therefore, requires capturing long-range temporal dependencies and aggregating information across entire sequences.

To address these challenges, we introduce **4DPC$^2$hat**, an MLLM for spatio-temporal reasoning over point cloud sequences, as illustrated in Fig. 1. To support the training, we first collect over 44K animated objects from the Objaverse series (Deitke et al., 2023b;a), and process them through a topology-consistent 4D point construction pipeline to obtain dynamic point cloud sequences, ensuring accurate pointwise temporal correspondence. We further employ a two-level captioning strategy, where the brief captioning emphasizes coarse alignment between holistic point cloud geometry and language, and detailed captioning focuses on fine-grained spatio-temporal semantics, including motion patterns, temporal evolution, and dynamic object states. This strategy enables the curation of over 200K high-quality question–answer pairs, supporting diverse question-answering (QA) tasks on appearance, counting, action recognition, temporal relations, and spatial reasoning.

To facilitate reasoning over long-range spatio-temporal patterns, we design an inter-frame bidirectional Mamba module that leverages state-space sequence modeling to capture dynamic behaviors among point cloud sequences while remaining efficient. However, we observe that naive supervised instructional finetuning (SFT) training with uniformly weighted data does not lead to balanced performance gains across diverse reasoning and perception abilities. We therefore introduce a failure-aware iterative bootstrapping learning strategy that systematically analyzes model failure cases and generates targeted QA pairs to expose weaknesses in specific domains. These targeted samples are incorporated into training, leading to progressively improved and more comprehensive model capabilities. We conduct extensive experiments to validate 4DPC$^2$hat and compare it with recent advanced MLLMs. The results show our substantial improvements in action understanding, temporal relation reasoning, and object interaction comprehension. Collectively, these gains indicate that 4DPC$^2$hat represents the first systematic effort towards enabling scalable and reliable reasoning over 4D dynamic point clouds, laying the groundwork for future advances in the field.

Our main contributions are summarized as follows:

- We introduce 4DPC$^2$hat, the first MLLM for 4D point clouds understanding, which effectively captures long-range irregular dependencies and dynamic behaviors, supporting diverse QA types.

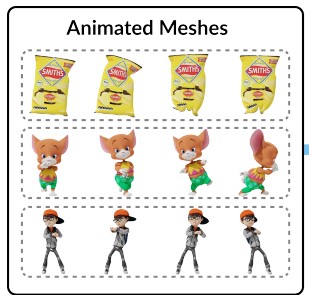 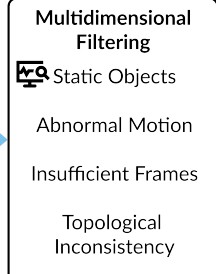 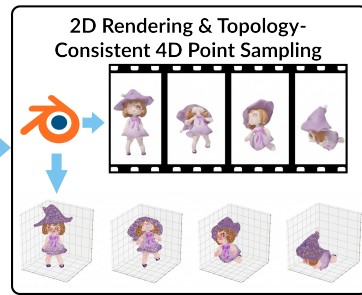

*Figure 2.* Illustration of the 4DPC²hat-200K dataset collection pipeline. The dataset contains 700K temporally ordered point cloud frames and 200K high-quality question–answer pairs, enabling both 4D object captioning and 4D object QA tasks.

*Table 1.* Comparison of existing datasets and ours (4DPC²hat-200K). Our dataset is the first 4D asset dataset to jointly support 4D object captioning and 4D object QA, featuring fine-grained diversity and over 200K QA pairs.

| Dataset | 4D Asset | Caption | Question-Answering | QA Diversity | Sample |
|---|---|---|---|---|---|
| PointLLM | ✗ | ✓ | ✓ | ✗ | 730K |
| PointLLMV2 | ✗ | ✓ | ✓ | ✗ | 1.8M |
| ShapeLLM | ✗ | ✓ | ✓ | ✗ | 860K |
| MiniGPT-3D | ✗ | ✓ | ✓ | ✗ | 730K |
| Diffusion4D | ✓ | ✗ | ✗ | ✗ | 82K |
| DeformingThings4D | ✓ | ✗ | ✗ | ✗ | 120K |
| **4DPC²hat-200K** | ✓ | ✓ | ✓ | ✓ | 200K |

- We propose a failure-aware bootstrapping learning pipeline that iteratively analyzes model deficiencies and generates targeted question–answering samples to progressively strengthen the corresponding abilities.

- We curate 4DPC²hat-200K, a large-scale cross-modal dataset comprising over 44K dynamic object point cloud sequences and more than 200K high-quality question–answer pairs for the community.

## 2. 4DPC²hat-200K Dataset

We systematically curate a large-scale dynamic point cloud understanding dataset, named 4DPC²hat-200K, by aggregating over 44k animated assets from Objaverse (Deitke et al., 2023b) and Objaverse-XL (Deitke et al., 2023a), as illustrated in Fig. 2. It is essential to guarantee the consistency of the dynamic topology for the whole sequence, as in Section 2.1, and important to provide both coarse and detailed descriptions (in Section 2.2), with diverse types of question-answering pairs to include multiple motion attributes (in Section 2.3). We compare our dataset with existing 3D and 4D datasets in Tab. 1. More details are in Appendix B.

### 2.1. Topology-Consistent 4D Point Construction

To ensure motion continuity and semantic completeness, animations with fewer than 16 frames are excluded, and long sequences are truncated to a maximum of 200 frames. For each asset, we employ an equidistant sampling strategy to extract $T = 16$ frames, striking an optimal balance between computational efficiency and temporal coverage of object dynamics. Furthermore, a lightweight motion-based filtering mechanism, based on inter-frame geometry differences, is applied to eliminate static or physically anomalous assets.

Then, the mesh assets are transformed into temporally consistent point cloud sequences using Poisson Sampling (Bridson, 2007). To ensure rigorous point-to-point correspondence across the frames, we only sample on the initial frame, where $N$ points are distributed proportionally to the surface area. Rather than re-sampling on each frame, we record the vertex indices and barycentric coordinates for each point of the first frame, and reconstruct their positions in subsequent frames by evaluating these coordinates against updated vertex positions. Noted that, we exclude sequences exhibiting topological changes and variations. Color attributes are assigned to the points and kept consistent. This yields a unified 4D representation of shape $(T, N, 6)$.

### 2.2. Two-level Captioning

After obtaining the rendered consistent image sequences from the chosen assets, we leverage Qwen2.5-VL (Bai et al., 2025) to generate two-level captions, including brief and complex captioning. The captions are also manually verified and corrected, such as for errors caused by obstructions. Specifically, brief captioning instructions require holistic descriptions of the dynamic point cloud, which are primarily used to facilitate latent space alignment between the point cloud encoder and the language model. Further more, complex captioning instructions, which demand richer and more detailed descriptions, including motion patterns and temporal evolutions across frames, are collected for fine-grained instruction fine-tuning. These designs enable the model to learn to generate temporally coherent and semantically expressive descriptions.

### 2.3. Question-Answering Generation

To construct diverse and structured question–answering pairs, we design QA instances from multiple complemen-

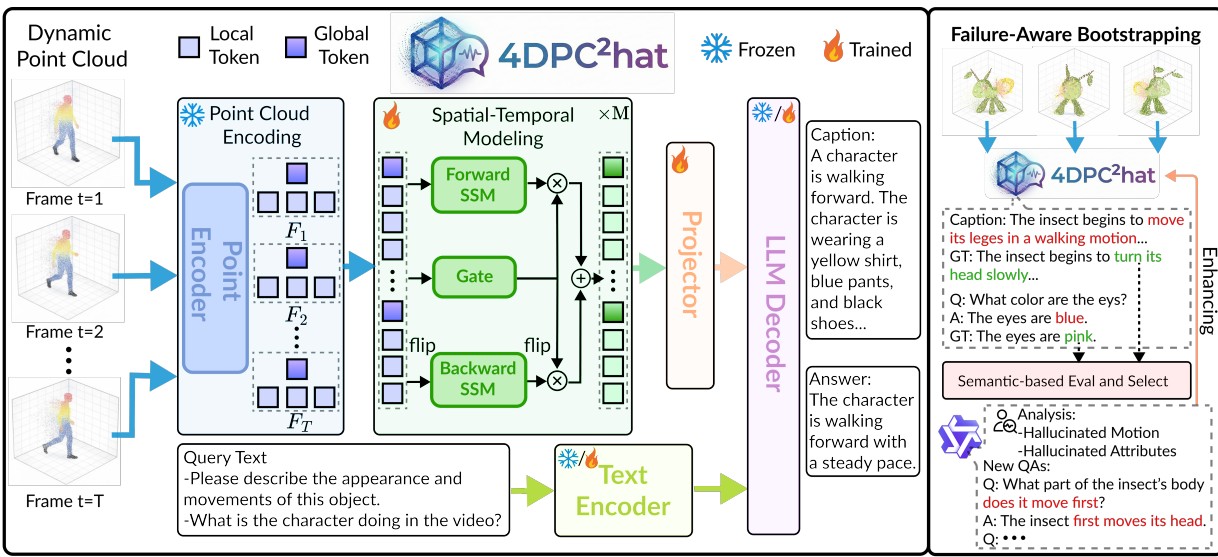

*Figure 3.* The 4DPC²hat framework. Dynamic point cloud frames are first encoded by Point-BERT into group-level and global tokens, followed by bidirectional Mamba-based temporal modeling across frames. The resulting spatio-temporal representation is aligned with LLM for 4D captioning and question-answering. Failure-Aware Bootstrapping Learning utilizes model's errors with semantic-based evaluation and selection, making analysis on failures and designing new QAs to further optimize the model on oriented, fine-grained data.

tary perspectives, including action, counting, appearance, temporal relationship, and spatial relationship. The QA pairs are generated based on the previously obtained complex dynamic captions, which are fed back into the LLM for QA generation. This yields high-quality QA pairs that are coherent with the captions.

## 2.4. Dataset Comparison and Diversity Analysis

As shown in Table 1, existing datasets either focus on static 3D captioning and QA (Xu et al., 2024; 2025a; Qi et al., 2024a; Tang et al., 2024) without modeling temporal dynamics, or provide 4D assets without language-centric annotations (Liang et al., 2024; Yang et al., 2021). Our dataset uniquely bridges this gap by jointly supporting 4D captioning and 4D QA at the asset level. Moreover, the QA annotations are designed with fine-grained diversity, enabling evaluation across diverse reasoning abilities. With over 200K QA pairs, our dataset strikes a trade-off among scale, temporal modeling, and diversity.

## 3. 4DPC²hat Pipeline

We illustrate the 4DPC²hat pipeline in Fig. 3. In Section 3.1, a designed efficient LLM adaptation architecture is introduced to capture fine-grained dynamic information for multimodal understanding. In Section 3.2, we explain the Failure-Aware bootstrapping learning strategy, utilizing the erroneous replies from the model to enhance its robustness on temporal reasoning. In Section 3.3, we describe our curriculum training process, from temporal feature alignment to failure-aware model refinement.

## 3.1. Efficient LLM Adaptation via Spatial-Temporal Modeling

4DPC²hat bridges 4D geometric information and LLM reasoning through a unified spatio-temporal architecture. We identify a "spatial over-compression" bottleneck in current temporal dynamic adaptation: aggregating frames into single global tokens discards localized motion cues and blurs action stages. Recognizing that 4D dynamics are inherently localized, our architecture retains multiple spatial group tokens alongside a global token per frame to preserve fine-grained fidelity. This unified sequence is modeled via a bidirectional Mamba module, enabling linear-complexity capture of long-range dependencies and symmetric temporal contexts essential for precise motion comprehension.

**Frame-wise Point Cloud Encoding.** Given a dynamic point cloud sequence $\mathcal{P} = \{P_1, P_2, \ldots, P_T\}, P_t \in \mathbb{R}^{N \times d}$, where $T$ denotes the number of frames, $N$ is the number of points per-frame and $d$ is the feature dimension of each point. Each frame $P_t$ is independently processed by a shared encoder $\mathcal{E}$ (Yu et al., 2022). Following the standard tokenization in Point-BERT (Yu et al., 2022), the point cloud is partitioned into $G$ local groups, each represented by a learnable group token. In addition, a global token is introduced for each frame to aggregate geometric context from all local tokens within one frame. The encoder $\mathcal{E}$ produces a set of tokens $F_t$ for each frame as follows:

$$F_t = \{f_{t,1}, f_{t,2}, \ldots, f_{t,G}, f_{t,\text{global}}\}, f_{t,g} \in \mathbb{R}^c, \quad (1)$$

where $G$ is the group number and $c$ is the feature dimension.

**Spatial-Temporal Modeling with Bidirectional Mamba.** As discussed before, we consider modeling the fine-grained 4D motions across different spatial regions using bidirectional Mamba (Zhu et al., 2024), for its long-range linear-complexity and capturing precise information. Given the obtained token sequence $F$ of all frames:

$$F = \{(f_{t,1}, f_{t,2}, \ldots, f_{t,G}, f_{t,\text{global}})\}_{t=1}^{T}, \quad (2)$$

where $f_{t,i}$ captures the localized dynamics, and $f_{t,\text{global}}$ encodes the overall motion to serve as an anchor.

Unlike Mamba (Gu & Dao, 2024), bidirectional Mamba (Zhu et al., 2024) enables linear-time sequence modeling while effectively capturing both forward ($F_f$) and backward ($F_b$) spatial-temporal context as follows:

$$F_f = \text{SSM}_f\big(\sigma(\text{MLP}_f(\text{LN}_f(F)))\big), \quad (3)$$

$$F_b = \text{flip}[\text{SSM}_b\big(\sigma(\text{MLP}_b(\text{flip}[\text{LN}_b(F)]))\big)], \quad (4)$$

where $\text{LN}(\cdot)$ denotes layer normalization, $\sigma(\cdot)$ is the GELU activation, and $\text{flip}[\cdot]$ reverses the token order along the sequence dimension. $\text{SSM}_f(\cdot)$ and $\text{SSM}_b(\cdot)$ denote forward and backward selective state-space operators. A following gating branch is applied to merge the forward and backward knowledge as follows:

$$F_g = \text{MLP}_1(\text{LN}_1(F)), \quad (5)$$

$$\tilde{F} = F + \text{MLP}_2((F_f \odot F_g + F_b \odot F_g)), \quad (6)$$

where $\odot$ is the pixel-wise dot product. We implement $K$ blocks to form the spatial-temporal bidirectional Mamba, which captures not only preceding motion cues but also how motions unfold and eventually terminate.

**Point Token Projection and Unified LLM Generation.** The enhanced feature $\tilde{F}$ is then projected into the language embedding space via a projection module $f_{\text{proj}}$:

$$F_{\text{proj}} = f_{\text{proj}}(\tilde{F}) \in \mathbb{R}^{T \times (G+1) \times c'}, \quad (7)$$

where $c'$ is the dimension of the hidden states of the LLM. The resulting point tokens are concatenated with text tokens to form a mixed token sequence, which is fed into a decoder-only LLM. The LLM then performs autoregressive generation conditioned on both geometric and temporal context, enabling open-ended captioning and question answering over dynamic point cloud sequences.

### 3.2. Failure-Aware Bootstrapping Learning Strategy

While supervised fine-tuning (SFT) on uniformly weighted data provides a baseline, it often fails to achieve balanced proficiency across heterogeneous spatio-temporal reasoning and perception scenarios. To enhance robustness in challenging scenarios, we propose an failure-aware bootstrapping learning strategy, as shown on the right of Fig. 3. The insight is that model failures serve as diagnostic signals, enabling targeted and iterative model refinement.

**Failure Identification and Selection.** Given a trained model $\mathcal{M}$ after SFT and a reference set $\mathcal{D}$, we perform large-scale inference to generate predictions $\hat{y} = \mathcal{M}(P, q)$. To quantify reasoning quality, we compute the semantic similarity $S$ between the prediction $\hat{y}$ and the ground-truth answer $y$ using a latent embedding space: $S(y, \hat{y}) = \frac{\phi(y) \cdot \phi(\hat{y})}{|\phi(y)||\phi(\hat{y})|}$, where $\phi(\cdot)$ denotes a pre-trained semantic encoder. We then rank the samples based on $S$ and isolate the bottom-performing $k\%$ as the failure set $\mathcal{D}_{\text{fail}}$, where the model exhibits significant spatio-temporal misunderstanding.

**Targeted Corrective Synthesis.** For each failure case in $\mathcal{D}_{\text{fail}}$, we employ a high-capacity teacher model Qwen-3 (Yang et al., 2025a) to synthesize corrective supervision. Specifically, we design a diagnostic prompt that guides the teacher model to categorize the error into one of 12 predefined taxonomies, and generate a new question-answer pair $(q', a')$ that directly probes the identified deficiency.

**Iterative Refinement.** After collecting the errors and generating new QA pairs, the model is progressively fine-tuned on the error-focused samples. This process is applied iteratively to rectify systematic biases, yielding an failure-aware model refinement. Experimental results demonstrate that this targeted approach yields substantially larger gains than naive data augmentation under comparable supervision budgets.

### 3.3. Curriculum Tuning and Refinement

**Temporal-Language Feature Alignment.** We first align dynamic point cloud representations with the LLM latent space to establish foundational temporal awareness. During this phase, the point cloud encoder and LLM are frozen, while only the bidirectional Mamba module and projector are optimized using 11k brief dynamic instructions. This alignment facilitates coarse-grained, distribution-level mapping across modalities.

**Comprehensive Instruction Tuning.** To enable complex reasoning and instruction following, we jointly fine-tune the projector, bidirectional Mamba module, and the LLM backbone. This stage utilizes 44k dynamic sequences paired with 145k question–answering pairs and 44k detailed captions, allowing the model to ground its linguistic responses in evolved geometric contexts. The point cloud encoder remains frozen to preserve stable geometric priors.

**Failure-Aware Refinement.** Finally, we apply the proposed failure-aware bootstrapping strategy to rectify systematic reasoning failures. To mitigate overfitting and catastrophic forgetting, the encoder and LLM are frozen, while

*Table 2.* 4D object captioning results on Objaverse. Results are reported using GPT-4 evaluation, semantic similarity metrics, and traditional language metrics. Compared with existing 3D-aware multimodal models, 4DPC²hat exhibits consistently robust performance across all metrics, demonstrating its effectiveness in modeling dynamic 4D object sequences. TA: Temporal Aggregation.

| Model | Input | GPT-4 | S-BERT | SimCSE | BLEU-1 | ROUGE-L | METEOR |
|---|---|---|---|---|---|---|---|
| PointLLM-7B (Xu et al., 2024) | 3D+TA | 47.98 | 50.21 | 46.48 | 15.59 | 15.08 | 11.27 |
| PointLLM-13B (Xu et al., 2024) | 3D+TA | 49.53 | 51.35 | 49.07 | 16.35 | 15.21 | 12.58 |
| ShapeLLM-7B (Qi et al., 2024a) | 3D+TA | 50.43 | 55.77 | 59.07 | 22.42 | 19.70 | 17.81 |
| ShapeLLM-13B (Qi et al., 2024a) | 3D+TA | 53.34 | 57.44 | 62.80 | 20.83 | 20.77 | 15.44 |
| MiniGPT-3D (Tang et al., 2024) | 3D+TA | 54.70 | 58.60 | 58.58 | 20.47 | 20.41 | 15.46 |
| **4DPC²hat** | **3D Point Cloud Sequence** | **73.27** | **79.08** | **82.03** | **38.40** | **43.31** | **36.29** |

*Table 3.* 4D object question answering on Objaverse. GPT-4 reports the average score over all question types, while category-specific (Counting, Temporal Relationship, Action, Spatial Relationship, and Appearance) results are evaluated using SimCSE. 4DPC²hat achieves consistently strong results across all question types.

| Model | GPT-4 | Counting | Temporal Relationship | Action | Spatial Relationship | Appearance |
|---|---|---|---|---|---|---|
| PointLLM-7B (Xu et al., 2024) | 52.69 | 51.27 | 58.22 | 55.34 | 62.36 | 50.91 |
| PointLLM-13B (Xu et al., 2024) | 54.15 | 57.21 | 58.78 | 51.07 | 62.48 | 56.71 |
| ShapeLLM-7B (Qi et al., 2024a) | 54.58 | 58.13 | 59.72 | 50.71 | 59.53 | 50.65 |
| ShapeLLM-13B (Qi et al., 2024a) | 56.17 | 56.95 | 60.48 | 52.32 | 61.64 | 52.38 |
| MiniGPT-3D (Tang et al., 2024) | 59.08 | 57.29 | 60.61 | 64.83 | 61.19 | 51.35 |
| **4DPC²hat** | **78.01** | **77.03** | **76.52** | **76.98** | **76.46** | **76.11** |

the Mamba module and the projector are refined on 12k targeted samples. This iterative process is applied twice. More details are in Appendix. C and D.

# 4. Experiments

## 4.1. Evaluation Metrics.

We evaluate model performance using a combination of traditional language metrics (BLEU-1 (Papineni et al., 2002), ROUGE-L (Lin, 2004), and METEOR (Satanjeev, 2005)), embedding-based semantic similarity (Sentence-BERT (Reimers & Gurevych, 2019) and SimCSE (Gao et al., 2021)), and LLM-based (GPT-4 (OpenAI, 2024)) judgment. We use $4,000$ object IDs as the test set for quantitative evaluation. Due to the high cost of GPT-4 inference, and following prior work (Xu et al., 2024; Qi et al., 2024a; Tang et al., 2024), GPT-4 evaluation is performed on a randomly selected subset of 200 object IDs.

## 4.2. Comparison with static 3D-aware MLLMs

We evaluate 4DPC²hat and compare it with representative static 3D-aware MLLMs, including PointLLM (Xu et al., 2024), ShapeLLM (Qi et al., 2024a), and MiniGPT-3D (Tang et al., 2024). To adapt these static methods to dynamic point clouds, we process each frame independently as a static input and aggregate frame-level predictions using a temporal summarization model (Qwen3) (Yang et al., 2025) to produce coherent dynamic-level captions. This protocol ensures a fair comparison between static 3D models

and our sequence-aware 4D approach. Detailed instructions are given in Appendix. F.

### 4.2.1. 4D OBJECT CAPTIONING

**Quantitative Comparison.** As Tab. 2 reports, 4DPC²hat significantly outperforms all adapted 3D-aware baselines on the 4D object captioning task. In particular, our model achieves a GPT-4 score of 73.27, surpassing the strongest baseline MiniGPT-3D (54.70) by 18.57 points. Similar trends are observed on embedding-based metrics (Sentence-BERT 79.08, SimCSE 82.03) and traditional captioning metrics (BLEU-1 38.40, ROUGE-L 43.31, METEOR 36.29). These large gains indicate that frame-wise processing followed by temporal aggregation is insufficient for generating coherent and temporally grounded captions. Our spatial-temporal modeling captures motion continuity and temporal dependencies absent in static 3D models.

**Qualitative Evaluation.** Fig. 4 compares captioning results on a dynamic 3D human sequence. PointLLM-13B focuses primarily on high-level appearance and potential functionality, while providing only vague references to motion. ShapeLLM-13B recognizes temporal evolution but describes it in abstract terms, failing to ground the motion into a specific action. MiniGPT-3D underestimates the dynamic content and incorrectly characterizes the sequence as largely static which contradicts the actual sequence. Overall, these static 3D models struggle to reliably infer dynamic behaviors from frame-wise aggregation, as fragmented temporal cues hinder the formation of coherent action-level semantics.

*Figure 4.* Qualitative comparison of 4D object captioning and QA across dynamic point cloud sequences. 4DPC$^2$hat demonstrates superior performance in action recognition, temporal reasoning, counting, appearance and spatial understanding compared to existing 3D-aware multimodal models, highlighting the advantage of directly modeling temporal dynamics from raw 4D point cloud sequences rather than frame-wise aggregation.

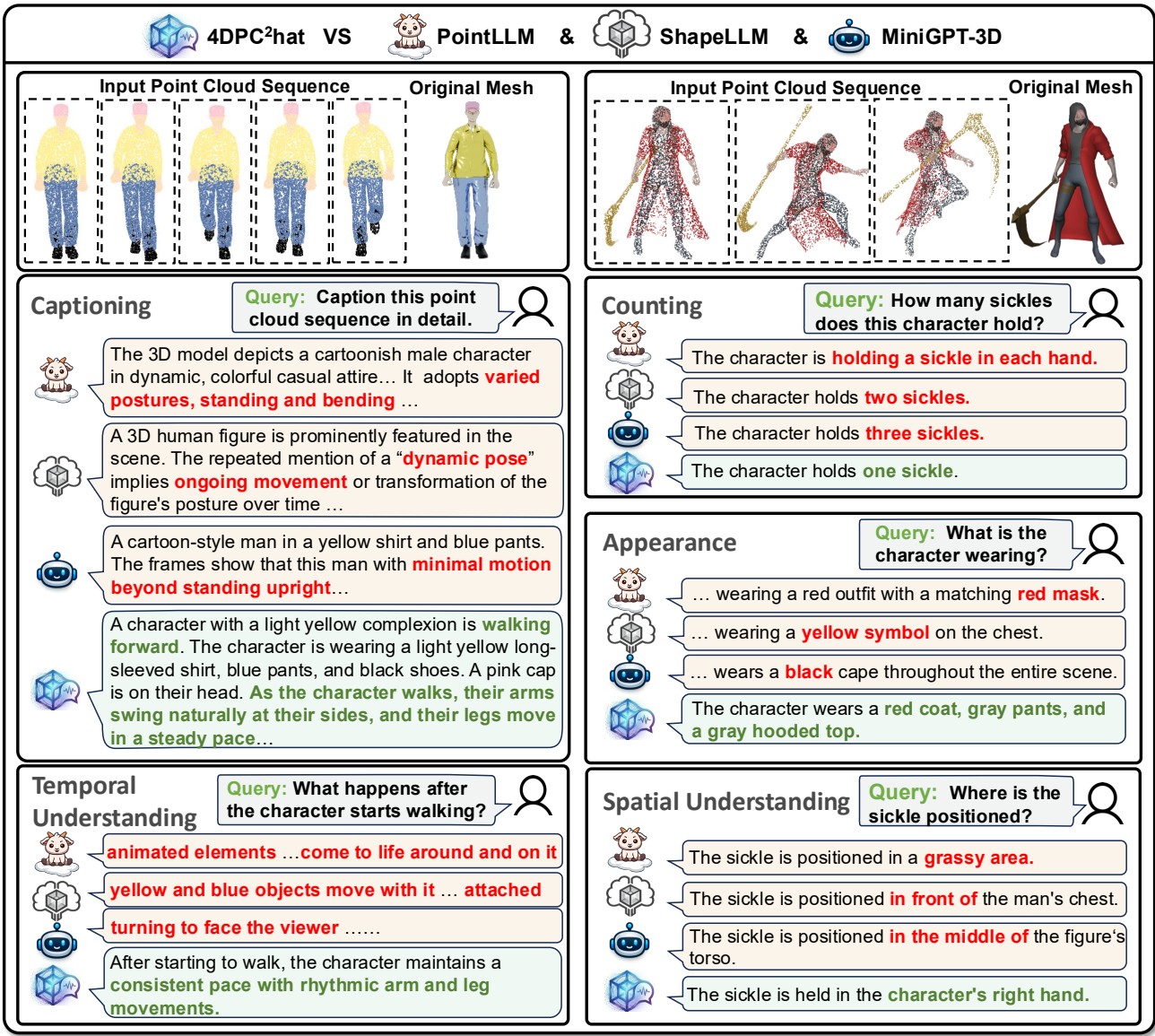

In contrast, our model accurately recognizes the walking action, clearly describes coordinated limb movements, and maintains consistent appearance across frames.

### 4.2.2. 4D OBJECT QUESTION ANSWERING

**Quantitative Comparison.** We further conduct qualitative evaluate for 4DPC$^2$hat on the 4D object QA task, covering five categories in Tab. 3. 4DPC$^2$hat consistently surpasses all baselines across question types, highlighting the effectiveness of temporal modeling for diverse 4D reasoning tasks. Overall, these results demonstrate that joint spatio-temporal reasoning over point cloud sequences is critical for reliable 4D question answering, enabling robust interpretation of actions, interactions, and state changes.

**Qualitative Evaluation.** Fig. 4 presents a qualitative comparison of representative model responses on the 4D object question answering task under identical prompts. In terms of temporal relationship, static baselines produce speculative descriptions involving unrelated objects or pose variations. 4DPC$^2$hat provides precise and consistent answers, cap-

turing the temporal ordering of object actions over time, reflecting stronger temporal reasoning and action-level understanding. Moreover, 3D models hallucinate color and appearance details and misinterpret spatial relationships from single-frame point clouds, leading to erroneous reasoning. In contrast, our model inputs point clouds across multiple frames, enabling much more accurate perception.

### 4.3. Comparison with 2D Video MLLMs

We conduct additional experiments to directly compare 2D Video MLLMs vs. our 4D point cloud-based model on the same tasks. The evaluation is on the open-source 4D-Bench (Zhu et al., 2025), which has already benchmarked a variety of MLLMs on multi-view rendered videos for 4D object understanding. We use the same objects' point cloud sequences as input.

#### 4.3.1. 4D OBJECT CAPTIONING

Existing 2D Video MLLMs mainly rely on temporally ordered 2D visual tokens for caption generation, making it difficult to jointly model dynamic geometry and temporal motion. As a result, these methods often struggle with fine-grained action evolution and viewpoint-consistent motion description. In contrast, our method directly models dynamic 4D point cloud sequences with explicit 3D geometry and motion information. As shown in Tab. 4, our model consistently outperforms existing 2D Video MLLMs on 4D object captioning, generating more accurate and temporally consistent action descriptions (e.g.,3.662/5 vs. 3.258/5).

#### 4.3.2. 4D OBJECT QUESTION ANSWERING

Reasoning over dynamic 4D objects is highly challenging for 2D Video MLLMs due to geometric ambiguity, occlusion, and cross-view inconsistency in 2D projections. Tasks such as counting, temporal reasoning, and motion understanding require accurate spatial-temporal perception that is difficult to achieve using only 2D visual tokens. By leveraging explicit 4D geometric representations, our framework naturally supports such reasoning. As shown in Tab. 5, our method achieves significant improvements over existing 2D Video MLLMs on 4D object question answering, particularly on action and counting tasks. Specifically, our model boosts action accuracy from 60.75% to 74.30% and object counting from 54.33% to 66.14%.

### 4.4. Ablation Study

**Transformer vs Mamba.** We compare a temporal Transformer with the proposed inter-frame temporal Mamba for temporal modeling in both 4D object captioning and question answering (Tab. 6 and Tab. 7). Inter-frame temporal Mamba consistently outperforms the Transformer across

*Table 4.* Comparison with 2D Video MLLMs on 4D Object Captioning (2D results from 4D-Bench)

| Model | METEOR | ROUGE | BERT | SBERT | GPT-appearance | GPT-action | GPT-eval |
|---|---|---|---|---|---|---|---|
| MiniGPT4-Video | 23.1 | 13.2 | 50.7 | 51.2 | 1.737 | 1.351 | 1.544 |
| InternVL2 8B | 27.9 | 22.6 | 58.2 | 60.3 | 2.531 | 1.877 | 2.204 |
| VideoChat2-Mistral | 33.5 | 33.5 | 65.4 | 59.7 | 2.578 | 1.912 | 2.245 |
| LLaVA-OneVision 7B | 39.2 | 32.7 | 63.2 | 65.6 | 3.166 | 2.479 | 2.823 |
| LLaVA-Video 7B | 41.7 | 38.8 | 66.7 | 68.1 | 3.235 | 2.552 | 2.894 |
| Qwen2-VL 7B | 36.9 | 36.4 | 65.7 | 66.9 | 3.170 | 2.666 | 2.918 |
| InternVL2 76B | 34.2 | 27.1 | 60.9 | 65.3 | 3.099 | 2.637 | 2.868 |
| LLaVA-OneVision 72B | 16.1 | 41.5 | 68.5 | 68.0 | 3.180 | 2.268 | 2.724 |
| LLaVA-Video 72B | 39.8 | 40.9 | 68.5 | 68.1 | 3.138 | 2.471 | 2.804 |
| Qwen2-VL 72B | 40.3 | 38.0 | 66.8 | 67.5 | 3.324 | 2.791 | 3.057 |
| Gemini 1.5 Flash | 36.5 | 32.9 | 65.3 | 68.9 | 3.246 | 2.931 | 3.088 |
| GPT-4o mini | 30.8 | 24.0 | 59.3 | 63.5 | 3.311 | 3.131 | 3.221 |
| Gemini 1.5 Pro | 38.7 | 39.0 | 68.5 | 68.8 | 3.311 | 2.983 | 3.147 |
| GPT-4o | 35.9 | 32.1 | 64.1 | 66.4 | 3.507 | 3.258 | 3.382 |
| 4DPC$^2$hat (Ours) | **42.9** | **43.4** | **70.1** | **73.2** | **3.794** | **3.662** | **3.728** |

*Table 5.* Comparison with 2D Video MLLMs on 4D Object Question Answering (2D results from 4D-Bench).

| Model | Object Counting (%) | Temporal Relationship (%) | Action (%) | Spatial Relationship (%) | Appearance (%) |
|---|---|---|---|---|---|
| MiniGPT4-Video | 22.05 | 26.43 | 22.90 | 22.39 | 22.06 |
| VideoChat2 | 22.83 | 31.43 | 33.18 | 38.81 | 34.56 |
| InternVL2 8B | 18.11 | 31.43 | 35.98 | 32.09 | 39.71 |
| LLaVA-OneVision 7B | 42.52 | 52.86 | 42.99 | 57.46 | 74.26 |
| LLaVA-Video 7B | 42.52 | 55.00 | 52.80 | 56.72 | 78.68 |
| Qwen2-VL 7B | 38.58 | 56.43 | 57.94 | 58.96 | 71.32 |
| InternVL2 76B | 28.35 | 45.00 | 42.52 | 38.81 | 64.71 |
| LLaVA-OneVision 72B | 49.61 | 58.57 | 60.75 | 61.19 | 76.47 |
| LLaVA-Video 72B | 54.33 | 58.57 | 57.48 | 66.42 | 77.21 |
| Qwen2-VL 72B | 45.67 | 55.71 | 58.41 | 61.19 | 72.06 |
| Gemini 1.5 Flash | 26.77 | 50.00 | 53.27 | 60.45 | 66.18 |
| GPT-4o mini | 40.16 | 50.71 | 50.00 | 61.94 | 72.06 |
| Gemini 1.5 Pro | 46.46 | 58.57 | 59.35 | 64.18 | 68.38 |
| GPT-4o | 44.09 | 59.29 | 63.55 | 69.40 | 77.21 |
| 4DPC$^2$hat(Ours) | **66.14** | **68.57** | **74.30** | **76.12** | **80.88** |

most metrics. This advantage stems from fundamental differences in temporal modeling. Temporal Transformers rely on global attention over sparsely sampled frames, which can smooth out subtle motion details and obscure fine-grained dynamics when temporal information is limited. In contrast, bidirectional Mamba performs sequential state-space modeling in both forward and backward directions, enabling more stable propagation of motion information and more effective integration of past and future context. As a result, dynamic patterns such as sustained actions, temporal transitions, and motion continuity are better preserved. Overall, this ablation demonstrates that bidirectional Mamba provides a more suitable inductive bias for long-horizon temporal modeling in dynamic point cloud sequences.

**Effect of Failure-aware Bootstrapping.** To isolate the contribution of our failure-aware bootstrapping learning strategy and verify that it specifically strengthens the model's weakest abilities, we start from the SFT baseline trained with uniform sampling and apply one and two rounds of bootstrapping refinement (Bs 1/Bs 2). This ablation targets our earlier observation that uniformly weighted SFT does not yield balanced gains across heterogeneous reasoning and perception skills, with temporal understanding being a clear bottleneck. As shown in Fig. 5 (b), bootstrapping improves the overall score monotonically and yields consistent gains across all question types. The largest improvements appear in the previously weaker categories, e.g., temporal reasoning improves from 71.41 to 76.52 and counting from 73.19 to 77.03 after two rounds. After the second iteration, the category scores become much more balanced, clustering around 76 and 77, the additional gains diminish, and certain categories show signs of saturation, so we stop bootstrapping at

*Table 6.* Temporal modeling ablation on 4D Object Captioning (higher is better). Transformer: Temporal Transformer, Mamba: Bidirectional Mamba.

| Temporal | GPT-4 | S-BERT | SimCSE | BLEU-1 | ROUGE-L | METEOR |
|---|---|---|---|---|---|---|
| Transformer | 69.08 | 77.98 | 79.03 | 37.67 | 41.78 | 35.34 |
| Mamba | **73.27** | **79.08** | **82.03** | **38.40** | **43.31** | **36.29** |

*Table 7.* Temporal modeling ablation on 4D Object QA.

| Temporal | GPT-4 | Counting | Temporal | Action | Spatial | Appearance |
|---|---|---|---|---|---|---|
| Transformer | 74.41 | 73.22 | 73.57 | 73.31 | 74.98 | 72.21 |
| Mamba | **78.01** | **77.03** | **76.52** | **76.98** | **76.46** | **76.11** |

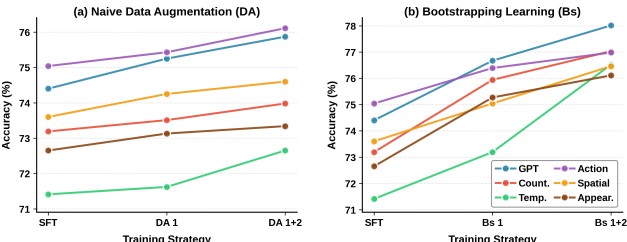

*Figure 5.* Merged comparison ablation on Naive Data Augmentation (DA) and Bootstrapping Learning (Bs).

two rounds.

**Bootstrapping Learning vs. Naive Data Augmentation.**
We further examine whether the gains mainly come from adding more data, by comparing bootstrapping to naive data augmentation under matched supervision budgets. In naive augmentation, we perform a single-stage SFT on the union of the original training data and the bootstrapped failure-targeted QA pairs, using one bootstrapping round for DA 1 and two rounds for DA 1+2. Fig. 5 (a) shows that this strategy yields limited and uneven improvements, with particularly small gains in temporal understanding and appearance. In contrast, bootstrapping uses the same base SFT model and then conducts a dedicated post-SFT refinement stage on failure-targeted QA pairs, concentrating optimization on underrepresented failure modes rather than diluting them in a uniformly mixed corpus. Consequently, bootstrapping achieves substantially larger and more balanced improvements: under GPT-4 evaluation, Bs 2 increases the overall score from 74.40 to 78.01, whereas DA 1+2 reaches 75.87. These results support our motivation that uniformly mixing augmented samples with the original corpus is insufficient for balanced capability gains, while targeted refinement on failure cases is markedly more effective.

**Influence of Bootstrapping Data Scale.** Fig. 6 illustrates the effect of increasing the amount of bootstrapping data on model performance. At each data scale, the first bootstrapping iteration provides the largest improvement, while the second iteration adds further but smaller gains, indicating diminishing returns from iterative refinement. Addition-

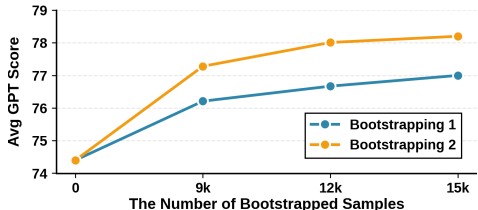

*Figure 6.* The Impact of Data Scale on Bootstrapping.

ally, as the data scale grows, especially from 12k to 15k samples, the marginal improvements become smaller, suggesting that the model has largely converged. This trend indicates that a moderate amount of bootstrapped data is sufficient to achieve most of the potential performance gains, while excessively increasing the data may lead to limited additional benefits. Therefore, we select 12k samples for our experiments, which provides a good balance between performance and data efficiency.

## 5. Conclusion

We introduced 4DPC$^2$hat, the first multimodal large language model specifically designed for understanding dynamic 4D object point cloud sequences. The proposed framework integrates a large-scale annotated 4D object point cloud dataset, bidirectional Mamba-based temporal modeling, and a failure-aware bootstrapping learning strategy to enable fine-grained action comprehension and temporal reasoning. Extensive experiments on 4D object captioning and question answering demonstrate consistent and substantial improvements over existing 3D-aware multimodal LLMs. By jointly advancing data, architecture, and training strategy, this work establishes a strong foundation for scalable and robust 4D object point cloud understanding. To further improve practical applicability, future work will focus on adapting the framework to real-world dynamic scenes and sensor data, such as LiDAR-based 4D perception.

## Acknowledgement

This work is supported by the National Natural Science Foundation of China (Grant No. 62502317), and the Guangdong Basic and Applied Basic Research Foundation (Grant No. 2026A1515011198).

## Impact Statement

This study introduces 4DPC$^2$hat, the first multimodal large language model (MLLM) designed for 4D point cloud reasoning, together with a large-scale dataset, 4DPC$^2$hat-200K, that supports high-level understanding tasks such as captioning and question answering. By overcoming the inherent limitations of static 3D perception, this research establishes

a foundation for future advances in 4D perception and embodied intelligence, with potential impact on robotics, simulation, and interactive AI systems.

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

# Appendix

## A. Related Work

**Multimodal Large Language Models.** Building on multi-modal LLMs, recent work (Zhu et al., 2023; Qi et al., 2024b; Hong et al., 2023b; Li et al., 2026b;a; 2025d;c; Fang et al., 2022; 2026; Zheng et al., 2022; 2023; Li et al., 2025a; Tan et al., 2025; Shi et al., 2025) has explored integrating vision with language understanding. Models such as PointLLM (Xu et al., 2024) and Point-LLMV2 (Xu et al., 2025a) enable language-driven reasoning over static 3D objects by aligning point cloud features with natural language, improving object recognition and spatial reasoning. ShapeLLM (Qi et al., 2024a) focuses on embodied interaction, providing universal 3D object understanding through a multi-view distillation-enhanced encoder.MiniGPT-3D (Tang et al., 2024) leverages 2D visual priors to improve 3D–language alignment, demonstrating that cross-modal knowledge can enhance instruction-following with limited 3D supervision. However, these models operate on single-frame or static point clouds. Their architectures and training data do not explicitly model temporal dynamics, limiting their ability to reason about motion, state changes, and object interactions over time. In contrast, our work moves beyond static perception to dynamic point cloud sequences, enabling temporal understanding and open-ended reasoning.

**4D Point Cloud Modeling.** Dynamic point cloud modeling focuses on learning spatio-temporal representations from point cloud sequences. Early works, such as FlowNet3D (Liu et al., 2019a), MeteorNet (Liu et al., 2019b), and PointRNN (Fan & Yang, 2019), primarily target short-term motion and local dynamics for tasks like scene flow estimation and simple action recognition. Later works by Hehe Fan et al. propose a series of methods (Fan et al., 2021b; 2023; 2021a), which aim to capture both local geometric structures and long-range temporal dependencies, achieving improved performance on action recognition and dynamic scene understanding. More recently, VG4D (Deng et al., 2024) extends 4D video understanding by combining point cloud sequences with vision-language modeling, enabling action recognition that leverages both spatio-temporal geometry and language priors. While these methods improve motion analysis and dynamic recognition, they are largely limited to specific tasks and cannot perform language-driven reasoning. Our framework enables unified spatio-temporal understanding and language-based reasoning from 3D point cloud sequences.

**Video Large Language Models.** Recent multimodal large language models (MLLMs) have achieved remarkable progress in video understanding (Shi et al., 2025; Xu et al., 2025b; Lee et al., 2026; Yang et al., 2025b; He et al., 2023; 2021; Sun et al., 2026; Yang et al., 2026; Diao et al., 2026; Hong, 2025; Li et al., 2025b). Early works such as VideoChat (Li et al., 2023) and MiniGPT4-Video (Ataallah et al., 2024) enable conversational video understanding through temporal feature alignment. General-purpose vision-language models, including InternVL (Chen et al., 2023), LLaVA-OneVision (Li et al., 2024a), and Qwen2-VL (Wang et al., 2024a), further improve video reasoning and multimodal perception. Proprietary systems such as GPT-4 Technical Report (Achiam et al., 2023) and Gemini 1.5 (Reid et al., 2024) also demonstrate strong long-context multimodal reasoning abilities. In addition, benchmarks such as MVBench (Li et al., 2024b) promote the evaluation of temporal reasoning and video understanding capabilities. Despite these advances, existing 2D video-based MLLMs still exhibit fundamental limitations when handling 4D object understanding tasks. Recent work 4D-Bench (Zhu et al., 2025) demonstrates that state-of-the-art video MLLMs struggle with fine-grained 4D object perception, particularly in tasks requiring spatial-temporal motion reasoning and viewpoint-consistent object representation. These limitations highlight the necessity of developing large multimodal models that can directly understand native 4D objects rather than relying on multi-view 2D video projections.

## B. Dataset Curation Details

**Convert to Point Cloud Sequence.** We implement an automated pipeline in Blender to convert 3D assets in GLB or FBX format into dynamic point cloud sequences. After importing a model, the scene is initialized and cleaned, and all mesh objects are parsed. If animation data are present, the valid animation frame range is automatically determined and truncated to a predefined maximum number of frames; otherwise, the asset is treated as a static object.

At the first frame, point sampling is performed on all mesh surfaces. The total number of points is distributed across meshes proportionally to their surface areas. For each mesh, points are sampled on triangular faces using barycentric coordinates, resulting in a fixed-size point cloud in the world coordinate system. Color attributes are assigned to each point following a priority order: vertex colors (if available), texture sampling, and material base color as a fallback. In addition to the 3D position and RGB color, we record the indices of the triangle vertices and the corresponding barycentric coordinates for each sampled point.

For subsequent frames, point positions are efficiently reconstructed by re-evaluating the stored barycentric coordinates with the updated vertex positions at each time step, avoiding repeated surface resampling. Color attributes are kept consistent across frames. From the full animation sequence, a fixed number of frames (e.g., 16) are uniformly sampled to form the final output, which is represented as a point cloud sequence of shape (T,N,6), containing (x,y,z,r,g,b) for each point. The resulting sequences are saved in a compressed format and used as input for downstream 4D perception and generation tasks.

**4D Object Captioning Annotation.** To generate captions, we employ Qwen-2.5-VL (Bai et al., 2025) as the annotation model. Following PointLLM (Xu et al., 2024), we employ a two-level captioning instruction design for dynamic point cloud sequences. We construct brief captioning instructions that require concise descriptions of the dynamic point cloud, primarily used to facilitate latent space alignment between point cloud features and language tokens. In addition, we include complex captioning instructions that demand richer and more detailed descriptions, explicitly involving motion patterns and temporal evolution across frames. These complex captions are used for instruction fine-tuning, encouraging the model to generate temporally coherent and semantically expressive descriptions. Example prompts are shown in the Fig. 1.

| 4D Object Detailed Captioning Prompt Template |
| --- |
| I provide you with 16 consecutive frames from a video. Your task is to generate one detailed caption in English for the scene. The caption should objectively describe the character's or object's physical appearance and narrate their movements in natural chronological order. Focus only on what can be directly observed, without interpretation, emotions, or symbolic descriptions. Do not mention the background. |

| 4D Object Brief Captioning Prompt Template |
| --- |
| The input is a detailed caption describing a short video clip. Your task is to generate a short and concise caption in English that only retains the key objects and actions. Remove any unnecessary details or modifiers. Keep the output to a single sentence if possible. |

*Figure 1.* Prompt for the multilingual large language model to generate detailed and brief descriptions of 4D objects. Within this prompt, we describe the object's actions, appearance, and changes over time.

**4D Object Question Answering Annotation.** To construct diverse and structured question–answer pairs for dynamic point cloud understanding, we design QA instances from multiple complementary perspectives, including Action, Counting, Appearance, Temporal Relationship, and Spatial Relationship.

All QA pairs are automatically generated using Qwen based on the previously produced detailed dynamic captions. Specifically, the detailed caption of each dynamic point cloud sequence is provided as input to Qwen, together with prompts (illustrated in the Fig. 3.), guiding the model to generate question–answer pairs that are strictly grounded in the caption content. This design ensures that each QA pair reflects observable properties and temporal behaviors described in the caption, without introducing external or speculative information. This process yields high-quality QA pairs with clear semantic focus and consistent alignment to the underlying dynamic point cloud content. Fig. 2 presents the category distribution for 4D object question-answering.

**Human verification of annotations.** For data construction, captions for the 44K point cloud sequences are first generated by Qwen2.5-VL and then manually verified and corrected. Each sample is reviewed by at least three invited annotators. We observe that Qwen2.5-VL tends to make errors in: object counting under occlusion and fine-grained action details. Human annotators correct these errors to ensure quality. QA pairs are derived from captions. We randomly sample 10% for manual verification and do not observe systematic issues.

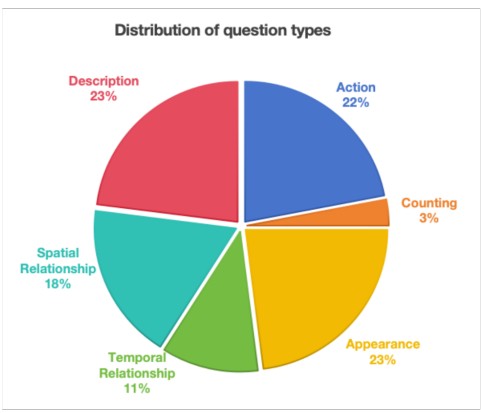

*Figure 2.* Distribution of the five subtasks within the 4D object question-answering task, with a total of 145k question-answer pairs.

## C. Data Construct from Bootstrapping Learning

We design a structured prompt to support failure-aware bootstrapping learning by explicitly diagnosing model failures and generating targeted corrective supervision. Given an object instance, the prompt compares the model output with the ground truth and classifies the failure into one or more predefined 4D reasoning error categories, covering motion, temporal order, spatial relations, interactions, counting, attention, and logical reasoning.

Based on the identified error type, the model generates exactly one new question–answer pair that directly probes the deficient reasoning capability. Strict constraints prevent paraphrasing, meta-questions, or hallucinated content, ensuring that all augmented questions are grounded in the ground truth while requiring deeper 4D understanding beyond the original query.

This prompt (Fig.4) enables scalable, fine-grained error analysis and targeted data augmentation, forming the core mechanism of our bootstrapping learning strategy.

## D. Experimental Settings

**Training Details.** We use LLaMA-7B (Touvron et al., 2023) model as the language backbone and a pretrained Point-BERT (Yu et al., 2022) model trained on Objaverse (Deitke et al., 2023b;a) as the point cloud encoder. Each point cloud contains $n = 8192$ points with $d = 6$ input dimensions. The encoder outputs $c = 384$ dimensional features per frame. For dynamic sequences, we uniformly sample $T = 16$ frames and feed the encoded features into the inter-frame temporal Mamba module to capture temporal dependencies. The temporally enhanced features are then projected into the LLaMA token space. All experiments are conducted on 8× NVIDIA A800 GPUs (80GB) using BF16 precision, FlashAttention, the AdamW optimizer, and a cosine learning rate scheduler.

**Three-stage training strategy.** Our training strategy consists of three phases that progressively shift the optimization focus from representation alignment to failure-aware reasoning refinement.

In the feature alignment phase, we use concise dynamic point cloud descriptions to bridge the representation gap between point cloud features and the language embedding space. This phase aims at coarse-grained, distribution-level alignment across modalities. The model is trained for 3 epochs using a batch size of 128 and a learning rate of 2e-3, with a total training time of approximately 4 hours.

Next, during instruction tuning, we train the model with detailed descriptions and multi-turn dialogues to enhance instruction-following and general reasoning capabilities. This phase is trained for 3 epochs with a batch size of 32 and a learning rate of 2e-5, taking approximately 12 hours to complete.

Finally, in the failure-aware model refinement phase, we focus on correcting systematic reasoning failures revealed during evaluation. Specifically, we evaluate the model on the 20% dataset, rank samples based on answer similarity to the ground truth, and select the lowest-performing 40% from the bottom of evaluated responses as failure cases. For each question-

---

**4D Object QA Prompt Template**

You are given a caption that describes a short video clip.
Your task is to design natural question–answer (QA) pairs based on the caption.

Generate one question-answer pair for each type below , provided that the description contains information of that type. If the caption does not contain information for a type, skip that type completely.
- Action: Recognize typical or subtle movements, or analyze directions of movement.
- Counting: Number of objects or specific features.
- Appearance: Object appearance attributes.
- Temporal Relationship: Order or sequence of actions/events over time.
- Spatial Relationship: Relative positions or arrangements of objects.

**Strict Rules:**
- For each type, you may generate at most one QA pair.
- The answer should not copy words from the caption directly. Instead, interpret the information and provide a natural, concise answer in your own words.
- The QA must be natural, human-like, concise, and strictly grounded in the caption.
- Keep questions and answers concise and avoid redundancy.
**Output Format (must be followed exactly):**
Q1 (Type): <question text>
A1: <answer text>

Q2 (Type): <question text>
A2: <answer text>

Q3 (Type): <question text>
A3: <answer text>

**Example:**
Caption: "A person wearing a red shirt waves their right hand, then sits down. There are three chairs and a table nearby."

Valid Output:
Q1 (Appearance): What is the person wearing?
A1: A red shirt.

Q2 (Action): What action does the person perform?
A2: They wave their right hand.

Q3 (Temporal Relationship): What happens after the wave?
A3: The person sits down.

Q4 (Counting): How many chairs are present?
A4: Three chairs.

Q5 (Spatial Relationship): Where is the table relative to the chairs?
A5: The table is nearby the chairs.

*Figure 3.* Prompts for the multilingual large language model to generate 4D question-answer pairs. Within this prompt, we examine five distinct perspectives to comprehensively formulate questions.

answer pair in every failure case, based on the identified error types, we generate targeted question–answer pairs that

---

**Bootstrapping Learning Data Generation Prompt Template**

You are a professional 4D visual reasoning data augmentation assistant.

You will receive one item containing:
- object_id
- question
- ground_truth
- model_output

Your task:
1. Carefully compare model_output and ground_truth.
2. Identify the real reason why the model_output is wrong.
3. From the 12 categories of 4D visual reasoning errors below, decide which error(s) apply:
(1) hallucinated motion
(2) missing motion
(3) motion type misclassification
(4) temporal order misunderstanding
(5) subtle detail ignorance
(6) spatial relation error
(7) interaction misunderstanding
(8) count misunderstanding
(9) hallucinated attributes
(10) pose mis-estimation
(11) attention failure
(12) reasoning logic failure

4. create 1-3 questions that:
- expose the exact type of error,
- require understanding the correct ground_truth,
- test the specific 4D reasoning capability the model failed,
- cannot be answered by memorizing or rephrasing the original question,
- are grounded strictly in the ground_truth (no invented content).

5. For each new_question, generate the correct new_answer based ONLY on ground_truth.

Strict constraints:
- ❌ Do NOT ask whether something is "correct", "accurate", "true", "valid", or "consistent".
- ❌ Do NOT ask "according to the description…".
- ❌ Do NOT mention "ground truth", "model output", or "the original question".
- ❌ Do NOT paraphrase or repeat the original question structure.
- ❌ No invented objects, actions, attributes, or events beyond ground_truth.

Output format:
Return ONLY a JSON LIST:

```
[
{
"new_question": "...",
"new_answer": "..."
}
]
```

No explanations.
No additional text.
No multiline answers.
"""

*Figure 4.* failure-aware prompt for Bootstrapping QAs generation.

emphasize temporal dynamics, motion attributes, and relational reasoning. To avoid disrupting the learned representation space, we freeze the Point-BERT encoder and the LLaMA backbone, and optimize only the temporal Mamba module and

---

**GPT-4 Metric Prompt Template**

Prompt Evaluate a model-generated caption against the ground truth for a 3D model. Identify the aspects mentioned in the ground truth and calculate the percentage of these aspects correctly mentioned or partially matched in the model caption. Score from 0 to 100, where each aspect contributes equally to the score. Consider similar concepts for a partial score.

Provide your score (0-100) and a short justification (less than 15 words) in the format of "score#reason"

Now score the following:

Human: {ground truth}

Model: {model output}

Output:

---

*Figure 5.* The prompt used in GPT-4 metric. GPT-4 compares scores how well the key information in the ground truth is correctly or partially reflected in the model output, producing a normalized score from 0 to 100.

the projection layers. This phase is trained for 3 epochs with a batch size of 32 and a learning rate of 2e-4. The bootstrapping learning is applied twice, with each iteration taking approximately 3 hours. These settings enable localized correction of failure cases while avoiding overfitting, catastrophic forgetting, or language drift.

## E. GPT Evaluation

We adopt GPT-4 as an auxiliary evaluator to assess the semantic alignment between model-generated captions and ground-truth descriptions for dynamic 3D objects. Given a predicted response and its corresponding reference caption, GPT-4 decomposes the ground-truth description into a set of semantic aspects and evaluates how many of these aspects are correctly or partially reflected in the model output. Each aspect contributes equally to the final score, resulting in a normalized evaluation score ranging from 0 to 100.

The exact evaluation prompt used in our experiments is shown in Fig. 5.

## F. Baseline Adaptation to Dynamic Point Cloud Sequences

Existing 3D-aware multimodal large language models, such as PointLLM, ShapeLLM, and MiniGPT-3D, are originally designed to operate on static point clouds and lack explicit mechanisms for temporal modeling. To enable a fair comparison in the 4D setting, we adapt these methods using a unified frame-wise evaluation protocol.

Specifically, each frame in a dynamic point cloud sequence is treated as an independent static input and processed by the baseline model to produce a frame-level textual response. These frame-level outputs are then aggregated into a single sequence-level prediction using a temporal consolidation module (Qwen3), which summarizes information across frames and produces a coherent description or answer for the entire sequence. This protocol allows static 3D models to be evaluated on dynamic inputs without modifying their architectures or introducing additional temporal supervision.

---

**Temporal Aggregation 3D Captioning Prompt Template**

The following text contains descriptions of individual frames in a dynamic scene.

{frame_text}

Do NOT summarize them frame by frame.
Do NOT mention frame numbers or temporal indices.

Instead, reason over the entire sequence and extract:
1. The key objects in the scene.
2. The dominant actions or motions that occur over time.
3. Any meaningful interaction or state change.

Merge repetitive information and describe the scene at the action/event level.

Your output should be a concise dynamic scene description (<100 words), not a list of frames.

Summary:

---

**Temporal Aggregation 3D QA Prompt Template**

You are given 16 static answers to the same question.
Each answer corresponds to one frame in a dynamic scene, representing how the question would be answered at that specific moment.
Please integrate all 16 frame-level answers to infer the overall or dynamic answer to the question.
Focus on the temporal progression and the general outcome across frames, rather than repeating static details from individual frames.

Provide a concise, coherent answer in one sentence (<30 words).

Question: {question}
Frame-level answers:{frame_text}
Final Answer:

---

*Figure 6.* Temporal aggregation prompt for enabling fair evaluation of static 3D models on dynamic point cloud sequences by summarizing frame-wise predictions with Qwen3.

## G. Additional Qualitative Results

We additionally provide qualitative comparisons between our model and representative static 3D models, as shown in Fig. 7. Furthermore, we present qualitative results of our model across different task categories in Fig. 8, illustrating its performance on diverse 4D reasoning scenarios.

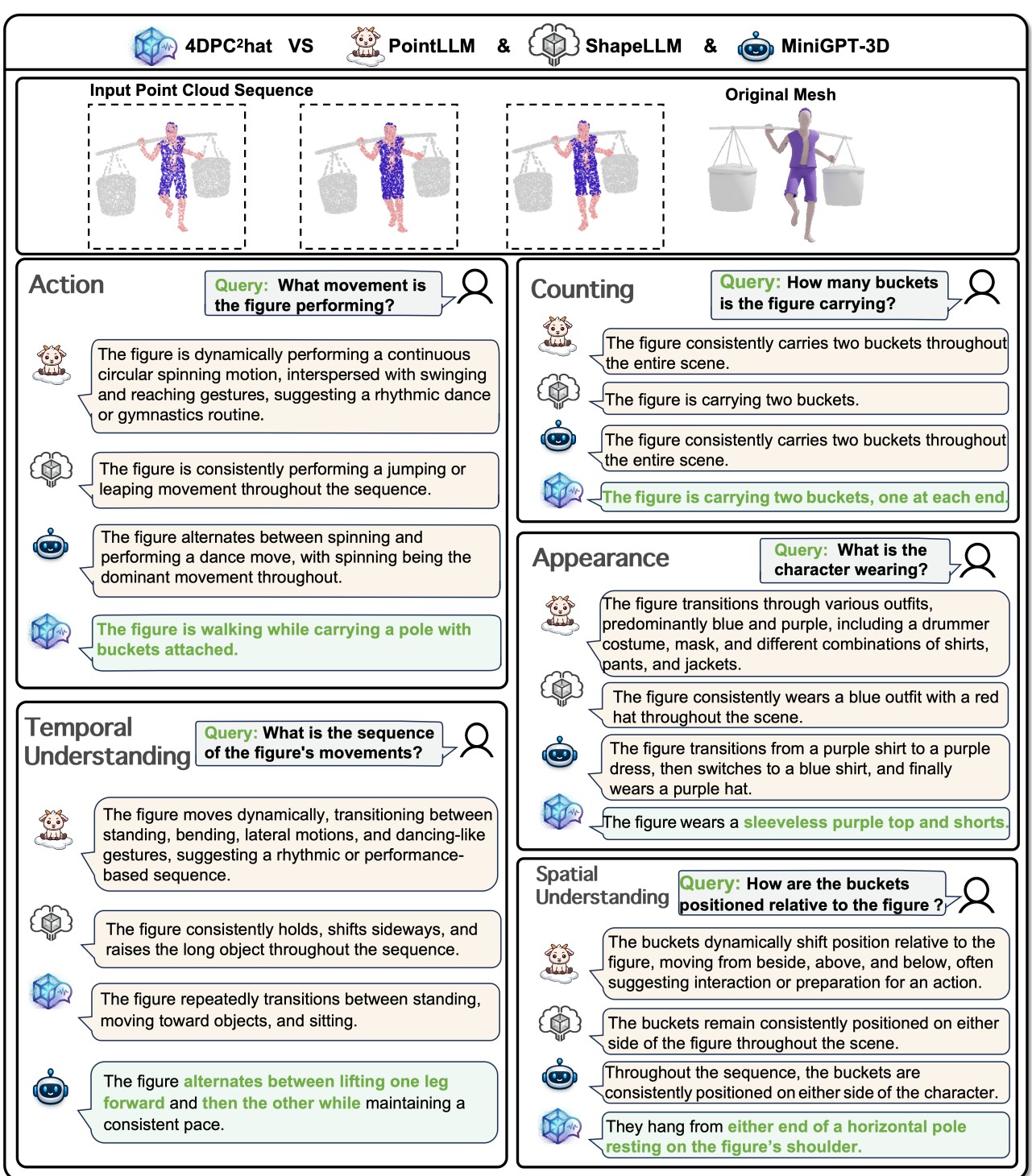

*Figure 7.* Qualitative comparison of 4D object captioning and QA across dynamic point cloud sequences.

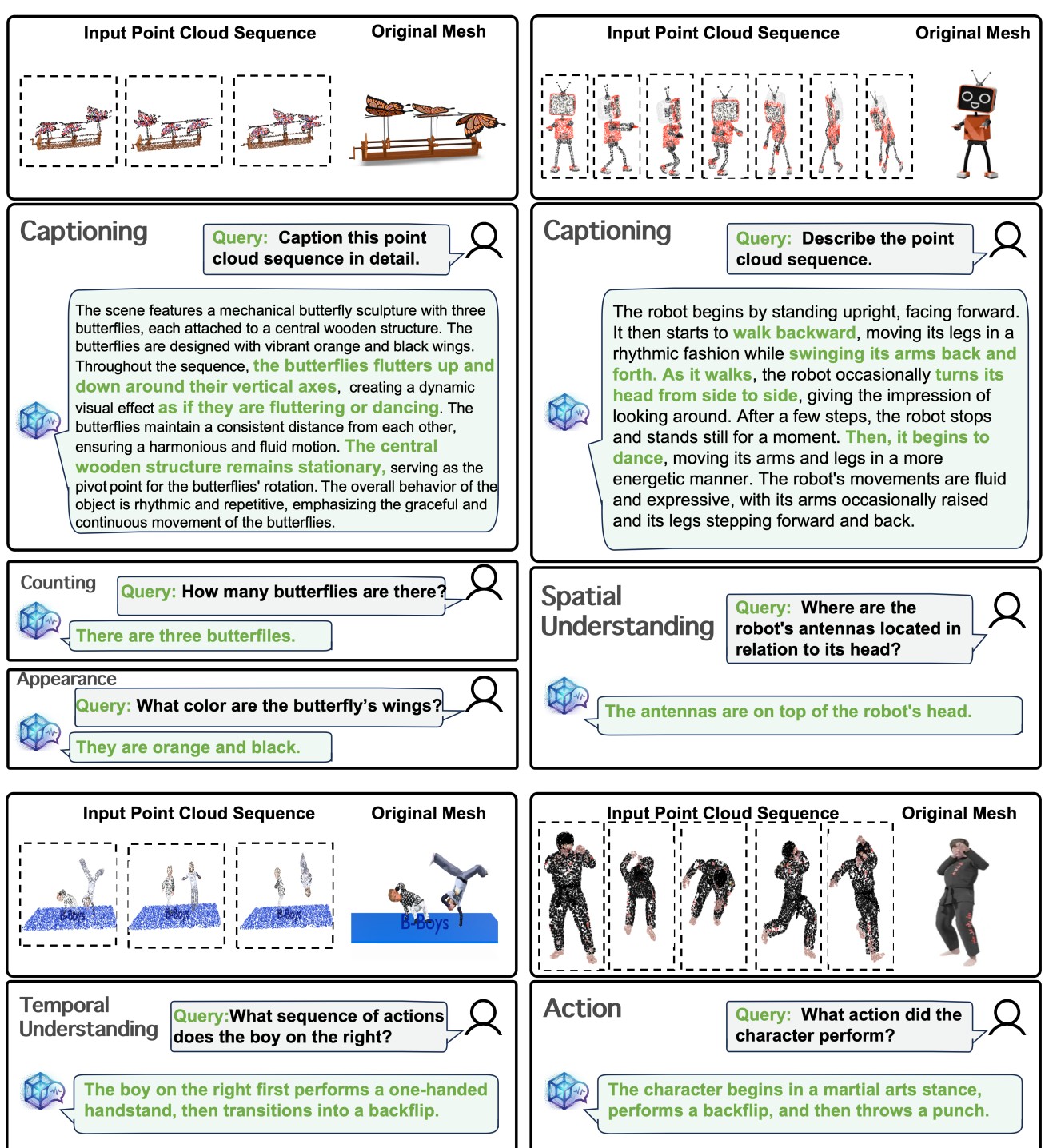

*Figure 8.* Results on 4D object captioning and QA across dynamic point cloud sequences.

