# OpenReview forum: "4DPC$^2$hat: Towards Dynamic Point Cloud Understanding with Failure-Aware Bootstrapping"
_ICML.cc/2026/Conference — ICML 2026 regular_

### Official Review · Reviewer_4VY5 · 2026-03-10

**Soundness:** 2
**Presentation:** 3
**Significance:** 3
**Originality:** 2
**Overall Recommendation:** 4
**Confidence:** 3

**Summary:**

This paper studies language understanding for dynamic object point clouds. The authors argue that existing 3D MLLMs mainly focus on static point clouds, lack cross-modal data for dynamic captioning and QA, and do not model temporal dynamics well. To address this, they build the 4DPC2hat-200K dataset from animated Objaverse assets and construct caption and QA supervision for topology-consistent 4D point cloud sequences. They also propose the 4DPC2hat framework, which keeps local and global tokens, uses bidirectional Mamba for cross-frame modeling, and connects the resulting features to an LLM for captioning and question answering. Experiments on 4D captioning and QA, together with ablations on temporal modeling and bootstrapping, show consistent gains over temporally adapted static 3D MLLM baselines.

**Compliance With Llm Reviewing Policy:**

Affirmed.

**Final Justification:**

The additional clarification improves the paper’s credibility. This reduces my concerns about data quality and evaluation robustness.

However, I do not think it fully remove the remaining limitations. such as that the real-world validation is still limited to 200 converted NTU RGB+D samples, and the strongest temporal baselines for direct compare are still not ideal.

Overall, I decide to keep the score.

**Key Questions For Authors:**

K1. Did you perform any human audit to verify the correctness, consistency, and major noise types of the automatically generated captions and QA pairs?

K2. Why are the main comparisons against frame-wise adaptations of static 3D MLLMs, rather than stronger dynamic temporal baselines or alternatives with capacity closer to your model?

K3. For failure-aware bootstrapping, could you provide sensitivity analysis for its key design choices?

**Limitations:**

No clear discussion about the methodological limitations.

**Strengths And Weaknesses:**

S1. The paper targets a clear and meaningful problem. It addresses the underexplored gap of “dynamic point clouds + language” in current 3D MLLMs, and clearly distinguishes the data and modeling needs of dynamic temporal understanding from those of static point cloud understanding.

S2. Beyond proposing a complete pipeline, the paper also contributes a potentially useful dataset for the community, which the authors state will be released in the future, although no explicit code release is mentioned.


S3. The model design is well motivated. It preserves local tokens and uses bidirectional Mamba to model cross-frame dynamics, rather than simply processing static 3D features frame by frame.

S4. Applying a failure-aware bootstrapping that focuses on failure cases rather than simply adding more data, and the paper compares it against naive augmentation.

S5. Experiments provides multiple perspective into this method, with main results, qualitative examples, and key ablations that support the main claims.


W1. The evidence mainly comes from the constructed Objaverse animation setting. Without validation on real sensor point clouds or clear cross-domain settings, generalization to more realistic 4D perception tasks remains unclear (section2 , section4)

W2. Both captions and QA annotations are automatically generated by LLM. It didn't provide human verification or error analysis. The reliability of the supervision is unclear ( section2.2, section2.3)

W3. GPT-4 evaluation is conducted on only 200 object IDs, with no variance or significance analysis, and the potential bias of using an LLM judge is not fully discussed (section4.1)

W4. The main comparisons are against frame-wise adaptations of static 3D MLLMs, not yet supports over a broader set of dynamic modeling alternatives (section4.2, Appendix F).

W5. The paper lacks systematic sensitivity analysis. (section3.2, section4.5)

---

> ### Author Rebuttal · Authors · 2026-03-31
>
> We thank the reviewer for the positive assessment, recognizing the motivation of the paper, the dataset contribution, the bidirectional Mamba-based temporal modeling, and the failure-aware bootstrapping strategy. We appreciate the reviewer’s suggestions and clarify the raised points below.
>
> **1. Validation on Real Sensor Point Clouds or Cross-domain Settings**
>
> To validate cross-domain generalization, we additionally evaluate on two real RGB-D datasets: NTU RGB+D-60/120 [1] by converting depth sequences into point cloud sequences and evaluate our model on 200 samples. Quantitative results and qualitative results are in the (https://anonymous.4open.science/r/icml-4B2A/rgbd2.png): We evaluate all QA categories, where the spatial relationship achieves the best performance (70.89), and other categories are around 60.
> While a performance drop is observed on real-world data due to domain shift, occlusion, and sensor noise, this is expected. Our method still maintains reasonable performance, indicating effective generalization on real sensor data and demonstrating cross-domain generalization ability.
>
> **2. Human Verification and Error Analysis**
>
> In MLLMs, it is common practice to use a data annotation pipeline that employs large language models as annotation tools, with final corrections made by human annotators. Due to space limitations, we omitted some details, which we clarify here. For data construction, captions for the 44K point cloud sequences are first generated by Qwen2.5-VL and then manually verified and corrected by three invited annotators. We observe that Qwen2.5-VL tends to make errors in: object counting under occlusion and fine-grained action details. Human annotators correct these errors to ensure quality. We randomly sample 10% QA pairs derived from captions for manual verification and do not observe systematic issues.
>
> **3. GPT-4 Evaluation on 200 Object IDs**
>
> First, GPT-4 is used in our paper as a complementary judge rather than the sole basis of the conclusions. Our main evaluation also reports conventional lexical and semantic metrics, and these are computed on the full 4000-object test set. Only the GPT-based evaluation is restricted to a randomly sampled subset of 200 object IDs due to API cost, which we will clarify more explicitly in the revision.
>
> Second, this setting is consistent with prior 3D point-cloud captioning practice. We follow prior works(e.g.,PointLLM, MiniGPT-3D, ShapeLLM).
>
> Third, we repeated the GPT-based evaluation 4 times under the same protocol. The 4D objects captioning score is [73.27±0.34], and the 4D object QA score is [78.01±0.51]. Our experimental results have consistently outperformed the strongest baseline.
>
> Fourth, to reduce dependence on a single LLM judge, we additionally conducted the same evaluation with Claude Sonnet 4.5 and Gemini 3 flash. Our method still consistently outperforms the baselines (https://anonymous.4open.science/r/icml-4B2A/llm.png). This provides further evidence that the improvement is not specific to one particular judge.
>
> **4. Comparison with Stronger Temporal Baselines**
>
> Existing dynamic point cloud methods mainly target discriminative tasks such as action recognition or segmentation, and they do not support open-ended multimodal generation for captioning and QA. Therefore, to the best of our knowledge, there is currently no off-the-shelf 4D point-cloud MLLM that is directly comparable to our setting. For this reason, we compare against the strongest feasible language-capable alternatives: (i) static 3D MLLMs adapted through frame-wise inference plus temporal aggregation, and (ii) 2D Video-LLMs on multi-view rendered videos (https://anonymous.4open.science/r/icml-4B2A/4d-table.png). These comparisons highlight the limitations of existing paradigms and further justify the necessity of explicit 4D point cloud modeling.
>
> **5. Sensitivity Analysis of Failure-aware Bootstrapping**
>
> We have already analyzed key design choices in Section 4.5 (Fig. 5 & Fig. 6) from three angles: (1) the number of refinement rounds (2) comparison with naive data augmentation under matched supervision budgets (3) the amount of bootstrapped data. The results show clear diminishing returns after 1-2 rounds. For example, temporal reasoning improves from 71.41 to 76.52 and counting from 73.19 to 77.03 after two rounds. Under matched supervision budgets, bootstrapping also outperforms naive augmentation (78.01 vs. 75.87 in GPT-4 evaluation). We further vary the number of bootstrapped samples and observe saturation beyond 12k, which is why we choose this value in the final configuration.
>
> These experiments systematically vary key parameters and evaluate performance, which constitutes a sensitivity analysis. We will revise the paper to explicitly highlight these results as a sensitivity analysis for improved clarity.
>
> [1]NTU RGB+D: A large scale dataset for 3d human activity analysis. CVPR, 2016.
>
> [2]Action recognition based on a bag of 3d points. CVPRW, 2010.

---

> > ### Author Rebuttal · Reviewer_4VY5 · 2026-04-03
> >
> > Most of my concerns have been addressed. I have two remaining questions:
> >
> > 1. For the newly added NTU RGB+D and 2D Video-LLM results, could the authors clarify the evaluation protocol more explicitly, including sample selection and input construction?
> >
> > 2. For data quality and evaluation robustness, the rebuttal mentions manual caption correction and a 10% QA audit. Could the authors further clarify the audit protocol, such as the sampling procedure, verification criteria, annotation process, and the approximate human effort required?

---

> > > ### Author Response · Authors · 2026-04-03
> > >
> > > We thank the reviewer for taking the time to carefully read our rebuttal and for recognizing most of our explanations. We appreciate the request for more details on the evaluation protocol and the audit protocol, and we clarify them as follows.
> > >
> > > 1.**Clarification of the Evaluation Protocol**
> > >
> > > **For 2D Video-LLM baselines**, we follow the open-source 4D-Bench [1], which benchmarks a variety of MLLMs on multi-view rendered videos for 4D object understanding. We use the same test objects and QA pairs provided by 4D-Bench, keeping the tasks identical (captioning and QA) to ensure a fair comparison. Specifically, we convert mesh assets (GLB/FBX) into dynamic point cloud sequences (NPZ format) following the pipeline described in our paper. The only difference between our method and the Video-LLM baselines lies in the input modality: Video-LLMs take multi-view rendered videos as input, while our model takes point cloud sequences. The results demonstrate that 4D point cloud representations provide clear advantages for 4D object understanding, highlighting the importance of our modality.
> > >
> > > **For NTU RGB+D**, we randomly sample 200 sequences from the NTU RGB+D 120 dataset. The dataset is collected from the real world and contains daily and mutual actions with aligned RGB videos and depth sequences. We map RGB values onto the depth data and convert them into point cloud sequences. To construct evaluation QA pairs, we prompt Qwen with RGB videos to generate diverse questions, all of which are manually verified. This setup enables evaluation on real-world RGB-D data under consistent task definitions.
> > >
> > > [1]4D-Bench: Benchmarking Multi-modal Large Language Models for 4D Object Understanding. ICCV,2025.
> > >
> > > 2.**Clarification of the Audit Protocol**
> > >
> > > **Sampling procedure.** For data construction, captions for the 44K point cloud sequences are first generated by Qwen2.5-VL, and then all manually verified and corrected. The refined captions are further used to construct QA pairs through qwen.  We randomly sample 10% of the QA pairs for additional human auditing.
> > >
> > > **Verification criteria.** The manual verification focuses on correcting errors commonly observed in 2D VLMs, such as counting under occlusion and fine-grained temporal dynamics, while ensuring consistency with the original animation.
> > >
> > > **Annotation process.** Each caption is reviewed by at least three annotators in a sequential manner. Annotators either approve or revise the caption, and the updated version is passed to the next annotator for further verification. For QA pairs, each sample is checked by at least one annotator.
> > >
> > > **Human effort.** The annotation process involves a professional trained team of 20 annotators. Each annotator reviews over 2,000 captions in the first pass, followed by second and third rounds of verification on samples previously reviewed by other annotators. Additionally, each annotator audits around 1,000 QA pairs. In total, the entire auditing process took approximately 4 weeks to complete.

---

### Official Review · Reviewer_dm1E · 2026-03-11

**Soundness:** 3
**Presentation:** 3
**Significance:** 3
**Originality:** 2
**Overall Recommendation:** 4
**Confidence:** 4

**Summary:**

This paper addresses dynamic 4D point cloud understanding, a setting that is not well handled by prior point-cloud MLLMs that mainly focus on static 3D inputs. It proposes 4DPC2hat, a multimodal model for dynamic point cloud sequences that combines point-cloud encoding with bidirectional Mamba-based temporal modeling, and further improves learning through a failure-aware bootstrapping strategy that generates targeted supervision from model errors. The paper also introduces 4DPC2hat-200K, a large-scale dataset containing over 44K animated assets, around 700K temporally ordered point cloud frames, and 200K QA pairs spanning captioning, action, counting, temporal, spatial, and appearance reasoning. Experiments show that the proposed framework substantially outperforms adapted 3D baselines on both 4D captioning and question answering, with especially strong gains in action understanding and temporal reasoning.

**Compliance With Llm Reviewing Policy:**

Affirmed.

**Final Justification:**

Main concerns have been well addressed.

**Key Questions For Authors:**

1.	Can the authors provide additional evidence of generalization beyond 4DPC2hat, either on an external dataset, a transferred downstream task, or a cross-benchmark evaluation setting? Since the current evaluation is largely conducted on the authors’ own benchmark, additional evidence of out-of-domain or cross-dataset generalization would significantly strengthen the paper’s claims.
2.	Can the authors provide a small-scale human verification study for the automatically generated captions and, if possible, for model outputs as well? The use of Qwen-based models for annotation and GPT-4 metrics for evaluation is understandable for scaling, but it also introduces possible annotation noise and judge bias. Even a modest manual audit of data quality and output quality would be informative.
3.	Can the authors clarify the train/validation/test split protocol in more detail, especially regarding near-duplicate animated assets, shared object families, or highly similar motion patterns across splits? Because the dataset is constructed from a large animated asset pool, it would be helpful to better understand how the split prevents overly similar instances from appearing across training and evaluation.

**Limitations:**

yes

**Strengths And Weaknesses:**

Strengths
1.	Existing point-cloud MLLMs mainly focus on static 3D inputs, while this work explicitly targets dynamic 4D point cloud understanding, which is highly relevant to embodied AI, robotics, and interactive perception. This makes the problem setting timely and meaningful.
2.	The dataset contribution is valuable and likely useful to the community. The paper curates 4DPC2hat-200K from over 44K animated assets, with 700K temporally ordered point cloud frames and 200K QA pairs, and supports both 4D captioning and 4D QA.
3.	The paper does not just claim that more data helps, but explicitly compares targeted bootstrapping with naive data augmentation under matched budgets. The results suggest the proposed refinement strategy yields larger and more balanced gains, especially on weaker capabilities such as temporal reasoning.

Weaknesses
1.	The evaluation is still largely confined to the authors’ own benchmark. Although the results on 4DPC2hat are strong, the paper would be more convincing with additional validation on external datasets or downstream tasks. As written, the evidence for generalization beyond the constructed benchmark is somewhat limited.
2.	The data construction and part of the evaluation rely heavily on automatic LLM pipelines. Captions and QA pairs are generated automatically using Qwen-based models, while evaluation also uses GPT-4 and embedding-based metrics. This is reasonable for scale, but it raises some concern about annotation noise, judge bias, and possible over-alignment to synthetic supervision. A small human verification study would strengthen the paper.

---

> ### Author Rebuttal · Authors · 2026-03-31
>
> We sincerely thank the reviewer for recognizing that our work addresses a timely and meaningful problem in dynamic 4D point cloud understanding, overcoming the limitations of prior point-cloud MLLMs that focus on static 3D inputs. We are also encouraged that the reviewer acknowledges the value of our 4DPC2hat-200K dataset and the effectiveness of our failure-aware bootstrapping strategy, particularly its advantage over naive data augmentation in achieving more balanced improvements.
>
> We address the concerns as follows.
>
> **1. Additional Evidence of Generalization on Other Benchmarks**
>
> To further validate the generalization ability of our model, we conducted two additional validations.
>
> First, we evaluate on 4D-Bench [1], an external benchmark containing 580 4D objects with diverse categories and high-quality annotations. We obtain the corresponding point cloud sequences and evaluate on the same tasks: 4D Captioning and 4D Question Answering (QA). As shown in Table 1 & 2 (https://anonymous.4open.science/r/icml-4B2A/4d-table.png), our model achieves consistent gains across multiple metrics on both tasks. Specifically, for 4D Captioning, our model improves GPT-action from 3.258 to 3.662 (+12.40%) and GPT-eval from 3.382 to 3.728 (+10.23%). For 4D QA, our model boosts Action accuracy from 60.75% to 74.30% (+22.30%) and Object Counting from 54.33% to 66.14% (+21.73%). Qualitative
> Results (https://anonymous.4open.science/r/icml-4B2A/4d-Qualitative.png) showing detailed and accurate descriptions. These results demonstrate that our model generalizes well to unseen benchmarks.
>
> Second, we further evaluate on real-world RGB-D datasets: NTU RGB+D 60 and NTU RGB+D 120 [2] by converting depth maps into point cloud sequences. On 200 samples, we show more quantitative results (https://anonymous.4open.science/r/icml-4B2A/rgbd2.png, Table 1), and qualitative results (https://anonymous.4open.science/r/icml-4B2A/rgbd2.png, Fig. 1). Our method still maintains reasonable performance, demonstrating effective generalization on real sensor data. We will add more cross-benchmark results in the final version.
>
> **2. Human Verification of Annotations and Model Outputs**
>
> In MLLMs, it is common practice to use a data annotation pipeline that employs large language models as annotation tools, with final corrections made by human annotators. Due to space limitations, we omitted some details, which we clarify here. For data construction, captions for the 44K point cloud sequences are first generated by Qwen2.5-VL and then manually verified and corrected. Each sample is reviewed by at least three invited annotators. We observe that Qwen2.5-VL tends to make errors in: object counting under occlusion and fine-grained action details. Human annotators correct these errors to ensure quality. QA pairs are derived from captions. We randomly sample 10% for manual verification and do not observe systematic issues.
>
> To reduce bias from a single LLM judge, we additionally conducted the same evaluation with Claude Sonnet 4.5 and Gemini 3 flash. The relative ranking remains consistent, with our method again outperforming the baselines (https://anonymous.4open.science/r/icml-4B2A/llm.png). This provides further evidence that the improvement is not specific to one particular judge.
>
> **3. Train/validation/test Split Protocol and Leakage Prevention**
>
> Our dataset is constructed based on object-level unique identifiers (UIDs) from Objaverse 1.0 and Objaverse-XL, so the same asset never appears across train/validation/test. This prevents data leakage and is a standard practice.
>
> Regarding shared object families or similar motion patterns, Importantly, Objaverse is large-scale and highly diverse. Moreover, our task goes beyond recognizing coarse motion patterns. And our task requires the model to generate fine-grained descriptions conditioned and answer conditioned jointly on spatial geometry and temporal dynamics. Even when two instances share a broad motion pattern, they still typically differ in geometric structure, articulated parts, object configuration, and the detailed way the motion unfolds over time.
>
> Thus, the model is hard to generate fine-grained descriptions by memorizing generic motion templates alone. Instead, it must learn meaningful instance-specific spatio-temporal reasoning. This interpretation is also supported by our transfer results on NTU RGB+D 60 and NTU RGB+D 120, where the model remains effective on real RGB-D point cloud sequences and unseen motion/object patterns. We will clarify this split protocol and its intended generalization scope more explicitly in the revision.
>
> [1]4D-Bench: Benchmarking Multi-modal Large Language Models for 4D Object Understanding. ICCV,2025.
>
> [2]NTU RGB+D: A large scale dataset for 3d human activity analysis. CVPR, 2016.

---

> > ### Author Rebuttal · Reviewer_dm1E · 2026-04-04
> >
> > The rebuttal has adequately addressed my main concerns. In particular, the additional cross-benchmark and transfer results provide meaningful evidence of generalization beyond the authors’ own benchmark, and the clarification on manual verification improves confidence in both the dataset construction and the evaluation setup. I therefore consider my concerns sufficiently resolved.
> > I will raise my score from 3 to 4.

---

> > > ### Author Response · Authors · 2026-04-04
> > >
> > > We thank the reviewer for acknowledging our work, for raising the score, and for the constructive follow-up. We are grateful that most of the concerns have been addressed, and will incorporate the additional experimental results into the final revision. We especially thank the reviewer for considering the overall value of the submission and for the supportive final assessment.

---

### Official Review · Reviewer_1GPj · 2026-03-12

**Soundness:** 3
**Presentation:** 3
**Significance:** 2
**Originality:** 2
**Overall Recommendation:** 3
**Confidence:** 3

**Summary:**

The paper proposes a 4D point cloud dynamics QA dataset, and a method trained on it. The paper shows the proposed method outperforms the baselines.

**Compliance With Llm Reviewing Policy:**

Affirmed.

**Final Justification:**

The authors solved some of the questions, but I'm still not fully convinced of the contribution of this dataset

**Key Questions For Authors:**

see weakness

**Limitations:**

see weakness

**Strengths And Weaknesses:**

* **Modality Necessity is Never Justified.**
This is the most fundamental flaw of the paper and alone warrants rejection. The paper never demonstrates that 4D point clouds are necessary for any of the proposed tasks. Examining every QA category — counting, appearance, spatial relationship, action, and temporal relationship — and every concrete example shown in Fig. 4, not a single one requires 3D information. "What is the character wearing?" is answered from a single image frame. "How many sickles does this character hold?" requires no temporal or depth information. "Where is the sickle positioned?" is a standard 2D spatial query. "What happens after the character starts walking?" is routine video understanding. Without exception, every demonstrated example is fully solvable by an RGB image or video model, rendering the entire point cloud pipeline unnecessary for the tasks as defined. This is not a minor gap — it means the paper has not established a valid use case for its core modality. The absence of image and video baselines is therefore not merely an incomplete ablation but a critical missing experiment that goes to the heart of the paper's validity. For such a comparison to be meaningful, image and video baselines must be fine-tuned on rendered image/video equivalents of 4DPC²hat-200K under matched supervision budgets — the same data fairness requirement the authors themselves failed to meet for their 3D baselines. If a fine-tuned video-LLM matches or outperforms 4DPC²hat, the entire contribution — dataset, architecture, and training strategy — is called into question, since all three are built around a modality that confers no demonstrated benefit. This concern is further compounded by the fact that the QA annotations were generated from rendered image sequences via Qwen2.5-VL, meaning the dataset is inherently image-biased by construction and may systematically disadvantage point cloud models relative to image-based ones. Fundamentally, the paper has designed a benchmark that does not require its own proposed modality, which undermines the validity of every experimental result reported.
* **Unfair and Potentially Uninformative Baseline Comparison.**
The paper trains 4DPC²hat on the proposed 4DPC²hat-200K dataset but adapts baseline models (PointLLM, ShapeLLM, MiniGPT-3D) purely at inference time via frame-wise processing and Qwen3 temporal aggregation, with no indication that these baselines are fine-tuned on the same dataset. Critically, the paper never explicitly states whether baselines were trained on 4DPC²hat-200K, which is itself a significant omission. If baselines are not fine-tuned on the proposed dataset — which the described adaptation protocol strongly suggests — then the observed performance gap (e.g., GPT-4 score of 73.27 vs. 54.70 for MiniGPT-3D) could be explained entirely by the in-distribution training advantage rather than any architectural superiority. This makes the comparison essentially meaningless: it pits a model trained in-distribution against models evaluated out-of-distribution. The correct experiment is to fine-tune all baselines on 4DPC²hat-200K and then compare, isolating architectural differences from data advantages. The authors must clarify this and provide properly controlled comparisons.
* **These Two Weaknesses Compound Each Other.**
Together, the missing modality justification and the unfair baseline comparison undermine the paper's core claims in a mutually reinforcing way. Even if the architectural comparison were made fair by fine-tuning baselines on the proposed dataset, a video-language model fine-tuned on the same data might outperform 4DPC²hat — simultaneously exposing both the comparison flaw and the questionable necessity of point clouds as a modality. Until both issues are addressed, the paper's central thesis — that 4D point cloud modeling with the proposed architecture provides meaningful benefits — is not demonstrated.
* **Evaluation Generalization.**
The test set is drawn from the same Objaverse distribution used for training, raising serious questions about generalization. There is no evaluation on held-out datasets or real-world dynamic scans such as LiDAR sequences. The heavy reliance on GPT-4 as an evaluator introduces additional circularity risks, since GPT-4 also informs the bootstrapping pipeline via Qwen3. A model that performs well under GPT-4 evaluation while being trained with GPT-4-family teacher models may be learning to satisfy the evaluator rather than genuinely improving at the underlying task.
* **Dataset Annotation Quality.**
The QA pairs are generated automatically from captions that were themselves automatically generated by Qwen2.5-VL from rendered images, creating a cascade of automatic annotation with compounding noise. No human evaluation of annotation quality is reported, making it difficult to assess the reliability of the supervision signal. More fundamentally, this pipeline means the dataset reflects what Qwen2.5-VL can observe from images — further reinforcing the concern that the benchmark is image-solvable by design rather than genuinely requiring 4D geometric reasoning.
* **Computational Cost of Bootstrapping.**
The bootstrapping pipeline requires large-scale inference, semantic similarity ranking, and teacher model synthesis at each iteration. No computational cost analysis is provided, making it unclear whether the gains over naive data augmentation justify the overhead in practice. This is particularly relevant given that the paper positions bootstrapping as a general and reusable strategy, yet provides no guidance on when it is cost-effective to apply.

---

> ### Author Rebuttal · Authors · 2026-03-31
>
> We thank the reviewer for the positive feedback such as “dataset contribution, state-of-the-art performance”, and the valuable suggestions. Below we make clarifications to each concerned point.
>
> **1. Modality Necessity**
>
> (1)4D point cloud is necessary for robust spatial-temporal reasoning. As shown in https://anonymous.4open.science/r/icml-4B2A/4d-Qualitative.png, image/video-based models fail due to the occlusion of dynamic objects in several views. The 4D point cloud indeed provides explicit, viewpoint-invariant geometry and motion in 3D space, regardless of occlusion, boosting robust understanding.
>
> (2)To better demonstrate, we conduct comprehensive comparisons with image/video-based models (input multi-view videos) on the open-source 4D-Bench [1]. The results can be found in https://anonymous.4open.science/r/icml-4B2A/4d-table.png. Although 4D-Bench is designed for multi-view video reasoning, our model significantly outperforms a series of 2D-based VLMs on diverse metrics and tasks. This verifies the necessity of our modality for robust and comprehensive temporal and spatial reasoning.
>
> (3)The dataset is inherently 3D-grounded, as Qwen2.5-VL serves only as an auxiliary drafting tool followed by rigorous human correction to rectify 2D hallucinations and ensure precise 3D spatial-temporal grounding, ensuring the QA pairs include structural truths beyond 2D visual features.
>
> (4)In addition,4D point clouds also offer practical advantages: (a)Privacy preservation: Sensitive visual details such as faces cannot be used as input. (b)Efficiency: Representing motion does not require dense video frames or multi-view rendering.
>
> **2. Fairness of Baseline Comparison**
>
> (1)We followed common practice in the 3D baselines(PointLLM, ShapeLLM, MiniGPT-3D), where they have not tuned their baselines on the proposed datasets, since (a) the dataset itself is a core contribution of this work; (b) the 3D baselines possess inherent architectural constraints, limiting their capability to model 4D temporal-spatial dynamics.
>
> (2)To isolate the impact of architecture from data advantage, we fine-tune baselines on our dataset. We evaluate them via frame-level captioning with GPT-based scoring: PointLLM-7B (45.82), ShapeLLM (49.35), and MiniGPT-3D (52.68), all showing poor performance. This is due to the limitation of static 3D models that lack temporal modeling. Even with external LLM aggregation, reliable temporal reasoning cannot be derived from high-error-rate inputs. The results confirm that the gains come from superior 4D modeling, rather than in-distribution training.
>
> **3. Generalization and Evaluation Robustness**
>
> (1)We validate cross-domain generalization on real-world NTU RGB-D-60/120 datasets [2]. The results shown in https://anonymous.4open.science/r/icml-4B2A/rgbd2.png, verify robust performance on unseen and real-world categories with sensor noise and occlusion.
>
> (2)To mitigate evaluation bias, GPT4 is used strictly as an independent metric, instead of a training signal, as evidenced by consistent gains in traditional metrics(BLEU/CIDEr). We additionally conducted
> the same evaluation with Claude Sonnet4.5 and Gemini 3 flash. Our method still consistently outperforms the baseline (https://anonymous.4open.science/r/icml-4B2A/llm.png).
>
> **4. Dataset Annotation Quality**
>
> (1)While utilizing MLLMs for QA and caption generation is standard practice, our pipeline incorporates intensive human-in-the-loop refinement: the 44K initial captions generated by Qwen2.5-VL are under rigorous manual correction by three independent annotators to rectify 2D-specific hallucinations.
>
> (2)This manual intervention specifically targets errors that 2D-VLMs struggle with, such as counting under occlusion and fine-grained temporal dynamics, injecting explicit geometric reasoning into the ground truth.
>
> (3)To ensure the reliability of the QA’s supervision, we manually audited 10% of the final QA pairs, which are generated from fully human-verified captions for logical consistency and 3D grounding, finding no significant issues; this confirms the benchmark targets structural truths rather than mere reflections of Qwen2.5-VL’s 2D observations.
>
> **5. Computational Cost of Bootstrapping**
>
> The computational overhead of our bootstrapping pipeline is highly manageable, with a one-time offline cost of inference on 10K samples taking 12 GPU-hours (a single GPU). The semantic similarity ranking is based on a lightweight CPU operation, which is negligible cost. The teacher model synthesis (LLM API call) at each iteration is applied only to failure cases (4k), and costs 6 hours, which is comparable to the computational cost for naive data augmentation. Importantly, this leads to consistent and non-trivial performance gain (+3.61) over naive data augmentation.
>
> [1]4D-Bench: Benchmarking Multi-modal Large Language Models for 4D Object Understanding. ICCV,2025.
>
> [2]NTU RGB+D: A large scale dataset for 3d human activity analysis. CVPR, 2016.

---

> > ### Author Rebuttal · Reviewer_1GPj · 2026-04-03
> >
> > will raise my score to 3.

---

> > > ### Author Response · Authors · 2026-04-04
> > >
> > > We sincerely thank the reviewer for the careful reading of our rebuttal, the constructive follow-up, and for acknowledging our work by raising the score. We truly appreciate the time and effort invested in reviewing our paper.
> > >
> > > We are grateful that most of the concerns have been addressed. We also appreciate the reviewer’s candid feedback in the initial round regarding the unclear motivation for introducing the modality. We respect this viewpoint and have addressed this concern in detail in the rebuttal. We will further clarify the purpose, significance, and necessity of the modality in the final revision, incorporating the additional experimental evidence described in the rebuttal.
> > >
> > > We also thank the reviewer for the positive feedback, including the recognition of our dataset contribution and state-of-the-art performance, as well as for the prompt response. If there are any remaining concerns, we would be happy to provide further clarification. Would the reviewer like us to further clarify any additional points?

---

### Official Review · Reviewer_4MkK · 2026-03-13

**Soundness:** 2
**Presentation:** 3
**Significance:** 2
**Originality:** 3
**Overall Recommendation:** 3
**Confidence:** 5

**Summary:**

The paper introduces $4DPC^{2}hat$, a novel multimodal large language model (MLLM) tailored for understanding dynamic 4D point cloud sequences. To address the lack of suitable training data, the authors introduce 4DPC2hat-200K, a dataset comprising 44K synthetic dynamic object sequences and 200K QA pairs derived from the Objaverse dataset. The proposed architecture employs an inter-frame bidirectional Mamba module to capture spatial-temporal dynamics. The authors also propose a failure-aware bootstrapping learning strategy to iteratively generate targeted QA pairs for refining the model's weaker capabilities. The model is evaluated against temporally aggregated static 3D MLLMs and shows superior performance on 4D object captioning and various QA tasks.

**Compliance With Llm Reviewing Policy:**

Affirmed.

**Key Questions For Authors:**

1. How does $4DPC^{2}hat$ compare to a modern 2D Video-LLM (e.g., Gemini, LLaVA-Video, GPT-4o) evaluated on multi-view 2D renderings of your test set? Could you provide empirical evidence demonstrating that the direct ingestion of 4D point clouds yields better reasoning or QA performance than 2D video understanding for the tasks proposed?
2. Can the authors demonstrate the model's performance on real-world dynamic point cloud data? Specifically, how does the model cope with background clutter, occlusions, and sensor noise typical of LiDAR or RGB-D captures, given that it was trained exclusively on clean, segmented Objaverse animations?

**Limitations:**

No. The authors have not adequately discussed the limitations of their work. The paper lacks a dedicated discussion regarding the model's exclusive reliance on synthetic, object-centric data and the inherent challenges of transferring this technology to noisy, real-world environments with background clutter and occlusions.

**Strengths And Weaknesses:**

**Strengths:**
1. The curation of the 4DPC2hat-200K dataset is a valuable effort, providing a much-needed resource for language-driven dynamic point cloud understanding.
2. Utilizing a bidirectional Mamba module for continuous spatial-temporal state-space modeling in 4D point clouds is an interesting and well-motivated alternative to temporal Transformers.
3. The failure-aware bootstrapping mechanism is an elegant, data-efficient approach to overcoming the uneven capability gains typical of standard supervised fine-tuning.

**Weaknesses:**
1. A fundamental flaw in the experimental design is the absence of any comparison with 2D Video-LLMs. Given the tasks evaluated (action recognition, counting, appearance, etc.), it is highly probable that state-of-the-art 2D Video-LLMs could achieve excellent results simply by processing rendered videos of these synthetic assets. Without this comparison, it is impossible to determine whether the explicit modeling of 4D geometry actually provides a tangible advantage over 2D video understanding for these specific QA tasks.
2. The model is trained and evaluated exclusively on clean, synthetic, object-centric animations. Real-world 4D point clouds (e.g., from LiDAR or RGB-D cameras) are characterized by severe background clutter, heavy occlusion, variable density, and sensor noise. The paper provides no experiments or discussions on how this architecture would perform outside of a perfect, synthetic vacuum, severely limiting its practical significance for Embodied AI or robotics.

---

> ### Author Rebuttal · Authors · 2026-03-28
>
> We sincerely thank the reviewer for the positive feedback on our work, including the recognition of the $4DPC^2hat-200K$ dataset, the effectiveness of our method, and its potential impact on advancing language-driven 4D point cloud understanding. We are especially encouraged that the reviewer highlights the value of our dataset and the proposed bidirectional Mamba module and failure-aware bootstrapping strategy. We illustrate the concerns as follows:
>
> **1. Comparison with Modern 2D Video-LLMs**
>
> We conduct additional experiments to directly compare **2D Video-LLMs vs. our 4D** point cloud-based model on the same tasks. The evaluation is on the open-source 4D-Bench [1], which has already benchmarked a variety of MLLMs on multi-view rendered videos for 4D object understanding. We use the same objects' point cloud sequences as input, keeping tasks identical (captioning and QA). Results can be found in the link (https://anonymous.4open.science/r/icml-4B2A/4d-table.png). In Tab.1 of captioning, our model produces more accurate and detailed action descriptions than 2D Video-LLMs (e.g., 3.662/5  vs. 3.258/5). Considering QA in Tab.2, we achieve significant improvements on counting, action recognition, and temporal relationship reasoning. We show more qualitative comparisons (https://anonymous.4open.science/r/icml-4B2A/4d-Qualitative.png) to confirm consistent improvements over other Video-LLMs.
>
> We attribute this to the fundamental limitations of 2D videos: multi-view videos require implicit fusion across viewpoints only in the RGB space, without any explicit geometric prior from the 4D point clouds, indicating the dynamics, spatial structure, and relationships.  This makes Video-LLMs' spatio-temporal reasoning more challenging and error-prone.
>
> **2. Experiments on Real-world 4D Point Clouds**
>
> (1) Considering the reason for training on synthetic data, we follow prior 3D LLM works (PointLLM (ECCV'2024), ShapeLLM (ECCV'2024), MiniGPT-3D (MM'2024) to build our dataset from Objaverse. The dataset is large and structurally complex, and contains information on object interactions. On the contrary, real-world 4D point cloud datasets with dense and high-quality language annotations are extremely scarce. Meanwhile, collecting aligned 4D point cloud sequences with caption data at scale is prohibitively expensive. This is the main reason we propose the first large-scale 4D point cloud multimodal understanding dataset, which itself is a key contribution.
>
> (2) Considering the generalization to real-world data,we further conduct experiments **on real-world NTU RGB+D 60/120 datasets** [2]. These datasets were collected from the real world and contain daily and mutual actions with depth sequences. We convert depth sequences into point cloud sequences and evaluate our model on 200 samples,  results in the (https://anonymous.4open.science/r/icml-4B2A/rgbd2.png) .
>
> The quantitative results are shown in Tab.1, where the spatial relationship achieves the best performance (70.89), and other categories are around 60. Although the RGB and depth data are not perfectly aligned, and there is unavoidable noise, our method still maintains reasonable performance, even outperforms PointLLM on the clean synthetic benchmark, demonstrating effective generalization on real sensor data.
>
> The qualitative results are displayed in Fig.1. Notably, even under occlusion and missing observations, our model can still achieve correct understanding. For example, in the third case, although the person’s right hand is occluded by the body over time, the model still correctly infers the action as hugging. In the first case, where writing and reading are often confused in 4D action classification [3], our model is able to correctly distinguish these two actions. These results indicate strong robustness and generalization ability on real-world data.
>
> (3) Considering the **LiDAR data**, we agree that LiDAR is common in the real world. However, existing LiDAR datasets lack aligned dynamic motion and dense language annotations. What’s more, the available LiDAR datasets (e.g., autonomous driving) are panoramic, with extremely sparse points and significant domain gaps. Prior works like PointLLM also mainly focus on 3D point clouds rather than LiDAR data. Emperically, we find that directly applying the baselines and our model to the LiDAR causes obvious performance degradation. As a result, directly transferring Lidar data for evaluation is non-trivial and requires dedicated preprocessing for domain adaptation.
>
> However, we will include a more detailed discussion in the final version and position our work as an important step toward dynamic 3D-language modeling, with future work focusing on real-world adaptation(e.g.,LiDAR).
>
> [1]4D-Bench: Benchmarking Multi-modal Large Language Models for 4D Object Understanding. ICCV,2025.
>
> [2]NTU RGB+D: A large scale dataset for 3d human activity analysis.CVPR,2016.
>
> [3]VG4D: Vision-Language Model Goes 4D Video Recognition.ICRA,2024.

---

### Decision · Program_Chairs · 2026-04-30

**Decision:**

Accept (regular)

**Comment:**

This paper received mixed reviews: two weak reject, two weak accept.

4MkK complained about missing comparison with 2D Video-LLMs, and missing evaluation in real-world non-object-centric scenes. 4MkK did not engage with the rebuttal, but the authors appear to have convincingly addressed the comparison with 2D video LLMs, and at least partially addressed the eval in real-world scenes with the use of 200 added NTU RGB+D samples, so the AC believes that if 4MkK re-engaged they would increase their score.

1GPj made a diverse set of complaints, and then based on the content of the rebuttal marked their review "Fully resolved". The final score is still only 'weak reject', but the AC takes this to mean borderline.

4VY5 identified a wide variety of strengths and wrote a positive review, and also the review increased in confidence after the rebuttal, though also notes that the 200-sample NTU eval is smaller than ideal.

dm1E also wrote a positive review, and seems especially impressed with the utility of the proposed dataset, and highlights the relevance of dynamic 3d scene understanding to today's ICML audience.

Taking these reviews in sum, the AC recommends to accept. The authors are encouraged to expand on the real-world non-object-centric evaluation, to resolve the main remaining weakness before the camera-ready.